# Association of genetic variation in *COL11A1* with adolescent idiopathic scoliosis

Hao Yu[1†], Anas M Khanshour[1†], Aki Ushiki[2,3†], Nao Otomo[4], Yoshinao Koike[4,5], Elisabet Einarsdottir[6], Yanhui Fan[7], Lilian Antunes[8], Yared H Kidane[1], Reuel Cornelia[1], Rory R Sheng[2,3], Yichi Zhang[2,3,9], Jimin Pei[10], Nick V Grishin[10], Bret M Evers[11,12], Jason Pui Yin Cheung[13], John A Herring[14,15], Chikashi Terao[5], You-qiang Song[7], Christina A Gurnett[8], Paul Gerdhem[16,17,18], Shiro Ikegawa[4], Jonathan J Rios[1,15,19,20], Nadav Ahituv[2,3], Carol A Wise[1,15,19,20]*

[1]Center for Translational Research, Scottish Rite for Children, Dallas, United States; [2]Department of Bioengineering and Therapeutic Sciences, University of California, San Francisco, San Francisco, United States; [3]Institute for Human Genetics, University of California, San Francisco, San Francisco, United States; [4]Laboratory of Bone and Joint Diseases, RIKEN Center for Integrative Medical Sciences, Tokyo, Japan; [5]Laboratory for Statistical and Translational Genetics, RIKEN Center for Integrative Medical Sciences, Yokohama, Japan; [6]Science for Life Laboratory, Department of Gene Technology, KTH-Royal Institute of Technology, Solna, Sweden; [7]School of Biomedical Sciences, The University of Hong Kong, Hong Kong SAR, China; [8]Department of Neurology, Washington University in St. Louis, St. Louis, United States; [9]School of Pharmaceutical Sciences, Tsinghua University, Beijing, China; [10]Department of Biophysics, University of Texas Southwestern Medical Center, Dallas, United States; [11]Department of Pathology, University of Texas Southwestern Medical Center, Dallas, United States; [12]Department of Ophthalmology, University of Texas Southwestern Medical Center, Dallas, United States; [13]Department of Orthopaedics and Traumatology LKS Faculty of Medicine, The University of Hong Kong, Hong Kong SAR, China; [14]Department of Orthopedic Surgery, Scottish Rite for Children, Dallas, United States; [15]Department of Orthopaedic Surgery, University of Texas Southwestern Medical Center, Dallas, United States; [16]Department of Surgical Sciences, Uppsala University, Uppsala, Sweden; [17]Department of Orthopaedics and Hand Surgery, Uppsala University Hospital, Uppsala, Sweden; [18]Department of Clinical Science, Intervention & Technology (CLINTEC), Karolinska Institutet, Stockholm, Uppsala University, Uppsala, Sweden; [19]Eugene McDermott Center for Human Growth and Development, University of Texas Southwestern Medical Center, Dallas, United States; [20]Department of Pediatrics, University of Texas Southwestern Medical Center, Dallas, United States

*For correspondence:
Carol.Wise@tsrh.org

[†]These authors contributed equally to this work

Competing interest: The authors declare that no competing interests exist.

**Abstract** Adolescent idiopathic scoliosis (AIS) is a common and progressive spinal deformity in children that exhibits striking sexual dimorphism, with girls at more than fivefold greater risk of severe disease compared to boys. Despite its medical impact, the molecular mechanisms that drive AIS are largely unknown. We previously defined a female-specific AIS genetic risk locus in an enhancer near the *PAX1* gene. Here, we sought to define the roles of *PAX1* and newly identified AIS-associated genes in the developmental mechanism of AIS. In a genetic study of 10,519 individuals

with AIS and 93,238 unaffected controls, significant association was identified with a variant in *COL11A1* encoding collagen (α1) XI (rs3753841; NM_080629.2_c.4004C>T; p.(Pro1335Leu); p=7.07E$^{-11}$, OR = 1.118). Using CRISPR mutagenesis we generated *Pax1* knockout mice (*Pax1*$^{-/-}$). In postnatal spines we found that PAX1 and collagen (α1) XI protein both localize within the intervertebral disc-vertebral junction region encompassing the growth plate, with less collagen (α1) XI detected in *Pax1*$^{-/-}$ spines compared to wild-type. By genetic targeting we found that wild-type *Col11a1* expression in costal chondrocytes suppresses expression of *Pax1* and of *Mmp3*, encoding the matrix metalloproteinase 3 enzyme implicated in matrix remodeling. However, the latter suppression was abrogated in the presence of the AIS-associated *COL11A1*$^{P1335L}$ mutant. Further, we found that either knockdown of the estrogen receptor gene *Esr2* or tamoxifen treatment significantly altered *Col11a1* and *Mmp3* expression in chondrocytes. We propose a new molecular model of AIS pathogenesis wherein genetic variation and estrogen signaling increase disease susceptibility by altering a PAX1-COL11a1-MMP3 signaling axis in spinal chondrocytes.

## eLife assessment

This **valuable** study analyzes a large cohort of Adolescent Idiopathic Scoliosis (AIS) patients, identifying an association with a variant in COL11A1 (Pro1335Leu). Experimental testing of this potentially pathogenic variant in vitro suggests a connection between Pax1, Col11a1, Mmp3, and estrogen signaling, thus providing **solid** support for the proposed link between hormonal and matrix components in the development of AIS.

## Introduction

The human spinal column is a dynamic, segmented, bony, and cartilaginous structure that is essential for integrating the brain and nervous system with the axial skeleton while simultaneously providing flexibility in three dimensions (*Richards et al., 2020*). Idiopathic scoliosis is the most common developmental disorder of the spine, typically appearing during the adolescent growth spurt. Adolescent idiopathic scoliosis (AIS) is reported in all major ancestral groups, with a population prevalence of 1.5–3% (*Wise, 2014*; *Hresko, 2013*). Children with AIS usually present with a characteristic right-thoracic major curve pattern and a compensatory lumbar curve. Major thoracolumbar and lumbar curves are less frequent (*Richards et al., 2020*). The three-dimensional nature of the deformity results in torsion in the spine that is most significant at the apex of the major curve, and changes in the structures of the vertebrae and ribs may develop as the curve worsens or progresses (*Richards et al., 2020*). Children with thoracic curves, with larger curves at first presentation, and/or with greater remaining growth potential are at increased risk of progression, but this risk decreases sharply after skeletal maturity (*Richards et al., 2020*). Sex is a recognized risk factor for AIS, with girls having at least a fivefold greater risk of progressive deformity requiring treatment compared to boys (*Karol et al., 1993*). This well-documented sexual dimorphism has prompted speculation that levels of circulating endocrine hormones, particularly estrogen, are important exposures in AIS susceptibility (*Liang et al., 2021*).

The genetic architecture of human AIS is complex, and underlying disease mechanisms remain uncertain. Heritability studies of Northern European (*Wynne-Davies, 1968*; *Grauers et al., 2012*), North American (*Riseborough and Wynne-Davies, 1973*; *Kruse et al., 2012*), and South Asian (*Tang et al., 2012*) ancestral groups suggest that disease risk is multifactorial, caused by genetic and environmental contributions (*Wise, 2014*; *Wise et al., 2020*). Accordingly, population-based genome-wide association studies (GWAS) in multiple ancestral groups have identified several AIS-associated susceptibility loci, mostly within non-coding genomic regions (*Wise et al., 2020*). In particular, multiple GWAS have implicated non-coding regions near the *LBX1* (*Takahashi et al., 2011*), *ADGRG6* (also known as *GRP126*) (*Kou et al., 2013*), and *BNC2* (*Ogura et al., 2015*) genes. An association with alleles in an enhancer distal to *PAX1*, encoding the transcription factor paired box 1, was primarily driven by females, suggesting that it contributes to the sexual dimorphism observed in AIS (*Sharma et al., 2015*). Subsequent meta-analysis of combined AIS GWAS identified additional susceptibility loci. These included variants in an intron of *SOX6*, a transcription factor, that along with *PAX1*, is important in early spinal column formation (*Smits and Lefebvre, 2003*). Furthermore, gene enrichment analyses found significant correlation of AIS-associated loci with biological pathways involving

**eLife digest** Adolescent idiopathic scoliosis (AIS) is a twisting deformity of the spine that occurs during periods of rapid growth in children worldwide. Children with severe cases of AIS require surgery to stop it from getting worse, presenting a significant financial burden to health systems and families.

Although AIS is known to cluster in families, its genetic causes and its inheritance pattern have remained elusive. Additionally, AIS is known to be more prevalent in females, a bias that has not been explained. Advances in techniques to study the genetics underlying diseases have revealed that certain variations that increase the risk of AIS affect cartilage and connective tissue. In humans, one such variation is near a gene called *Pax1*, and it is female-specific.

The extracellular matrix is a network of proteins and other molecules in the space between cells that help connect tissues together, and it is particularly important in cartilage and other connective tissues. One of the main components of the extracellular matrix is collagen. Yu, Kanshour, Ushiki et al. hypothesized that changes in the extracellular matrix could affect the cartilage and connective tissues of the spine, leading to AIS.

To show this, the scientists screened over 100,000 individuals and found that AIS is associated with variants in two genes coding for extracellular matrix proteins. One of these variants was found in a gene called *Col11a1*, which codes for one of the proteins that makes up collagen.

To understand the relationship between *Pax1* and *Col11a1*, Yu, Kanshour, Ushiki et al. genetically modified mice so that they would lack the *Pax1* gene. In these mice, the activation of *Col11a1* was reduced in the mouse spine. They also found that the form of *Col11a1* associated with AIS could not suppress the activation of a gene called *Mmp3* in mouse cartilage cells as effectively as unmutated *Col11a1*. Going one step further, the researchers found that lowering the levels of an estrogen receptor altered the activation patterns of *Pax1*, *Col11a1*, and *Mmp3* in mouse cartilage cells. These findings suggest a possible mechanism for AIS, particularly in females.

The findings of Yu, Kanshour, Ushiki et al. highlight that cartilage cells in the spine are particularly relevant in AIS. The results also point to specific molecules within the extracellular matrix as important for maintaining proper alignment in the spine when children are growing rapidly. This information may guide future therapies aimed at maintaining healthy spinal cells in adolescent children, particularly girls.

cartilage and connective tissue development (*Khanshour et al., 2018*). A more recent GWAS in a Japanese population identified 14 additional AIS loci that are candidates for further evaluation (*Kou et al., 2019*). In separate studies, genome sequencing in AIS cases and families identified enrichment of rare variants in the *COL11A2* (*Haller et al., 2016*) and *HSPG2* (*Baschal et al., 2014*) genes, encoding components of the cartilage extracellular matrix (ECM). Hence, variation affecting cartilage and connective tissue ECM is an emerging theme in the heterogeneous genetic architecture of AIS.

Pre-clinical animal models are essential tools for accelerating mechanistic understanding of AIS and for therapeutic testing (*Wise et al., 2020*). In zebrafish, several genetic mutants with larval or later-onset spinal deformity have been described, including *ptk7* (*Hayes et al., 2014*; *Van Gennip et al., 2018*), *c21orf59* (*Jaffe et al., 2016*), *ccdc40* (*Becker-Heck et al., 2011*), *ccdc151* (*Bachmann-Gagescu et al., 2011*), *dyx1c1*, and *kif6* (*Konjikusic et al., 2018*). In rescue experiments, Rebello et al. recently showed that missense variants in *COL11A2* associated with human congenital scoliosis fail to rescue a vertebral malformation phenotype in a zebrafish *col11a2* knockout line (*Rebello et al., 2023*). In mouse, conditional deletion of *Adgrg6* in skeletal cartilage (using *Col2a1*-Cre) produces a progressive scoliosis of the thoracic spine during postnatal development that is marked by herniations within the cartilaginous endplates of involved vertebrae. Progressive scoliosis, albeit to a lesser extent, was also observed when *Adgrg6* was deleted from committed chondrocytes (using ATC:Cre) (*Long et al., 2001*; *Liu et al., 2019*; *Liu et al., 2021*). These studies demonstrate that cartilage and possibly other osteochondroprogenitor cells contribute to the scoliosis phenotype in these models (*Liu et al., 2019*). Taken together, genetic and functional studies in mouse, although limited, support the hypothesis that deficiencies in biogenesis and/or homeostasis of cartilage, intervertebral disc (IVD), and dense connective tissues undermine the maintenance of proper spinal alignment during the adolescent growth spurt (*Wise et al., 2020*).

**Table 1.** Study cohorts.

| Cohort | Ethnicity | Stage | Subjects | Cases | | Controls | |
|---|---|---|---|---|---|---|---|
| | | | | Male | Female | Male | Female |
| USA (TX) | NHW | Discovery | 13,865 | 201 | 1157 | 5369 | 7138 |
| USA (MO) | NHW | Replication | 2951 | 201 | 1102 | 1049 | 689 |
| SW-D | NHW | Replication | 4627 | 222 | 1409 | 505 | 2491 |
| JP | EAS (Japanese) | Replication | 79,211 | 323 | 5004 | 40,205 | 33,679 |
| HK | EAS (HAN Chinese) | Replication | 3103 | 178 | 812 | 858 | 1255 |
| Total | | | 103,757 | 10,519 | | 93,238 | |

USA (TX): Texas cohort; USA (MO): Missouri cohort; SW-D: Danish cohort; JP: Japanese cohort; HK: Hong Kong cohort; NHW: Non-Hispanic White; EAS: East Asian.

The combined contribution of reported AIS-associated variants is broadly estimated to account for less than 10% of the overall genetic risk of the disease (**Kou et al., 2019**). To address this knowledge gap, we sought to define novel loci associated with AIS susceptibility in genes encoding proteins of the ECM (i.e. the 'matrisome'; **Naba et al., 2012b**; **Naba et al., 2012a**). Here, we identify new genetic associations with AIS. Further, our functional assessments support a new disease model wherein AIS-associated genetic variation and estrogen signaling perturb a PAX1-COL11a1-MMP3 axis in chondrocytes.

## Results
### Nonsynonymous variants in matrisome genes are associated with increased risk of AIS

The 'matrisome' has been defined as 'all genes encoding structural ECM components and those encoding proteins that may interact with or remodel the ECM' (**Hynes and Naba, 2012**). Proteins comprising the global ECM as currently defined have been identified by both experimental and bioinformatic methods (**Naba et al., 2012b**). We assembled 1027 matrisome genes as previously identified (**Naba et al., 2016**), including 274 core-matrisome and 753 matrisome-associated genes (N=1027 total). For the genes encoding these 1027 proteins, we identified all nonsynonymous common variants (MAF>0.01) queried by the Illumina HumanCoreExome-24v1.0 beadchip and determined their genotypes in a discovery cohort of 1358 cases and 12,507 controls, each of European ancestry (**Table 1**). After applying multiple quality control measures (see Methods section), we retained 2008 variants in 597 matrisome genes for association testing (**Supplementary file 1**). This sample size was estimated to provide at least 80% power to detect significant associations at the matrisome-wide level ($\alpha \leq 2.5E^{-05}$), for alleles with population frequency $\geq 0.05$ and OR $\geq 1.5$ (**Figure 1—figure supplement 1**). Two nonsynonymous variants, in *COL11A1* (rs3753841; NM_080629.2_c.4004C>T; p.(Pro1335Leu); odds ratio (OR)=1.236 [95% CI = 1.134–1.347], p=1.17E$^{-06}$) and *MMP14* (rs1042704; NM_004995.4_c.817G>A; p.(Asp273Asn); OR = 1.239 [95% CI = 1.125–1.363], p=1.89E$^{-05}$) were significantly associated with AIS (**Figure 1A**). Given the sexual dimorphism in AIS and our prior observation of a female-predominant disease locus (**Sharma et al., 2015**), we tested the 2008 variants separately in females (N=1157 cases and 7138 controls). In females, the association with rs3753841 remained statistically significant, whereas rs1042704, near *MMP14*, was not associated with AIS in females (**Figure 1—figure supplement 2**). Our study was not sufficiently powered to test males separately.

To validate these results, we sought to replicate the associations of rs3753841 and rs1042704 in four independent AIS case-control cohorts, from North America, Europe, and eastern Asia, representing multiple ethnicities (total N=9161 AIS cases, 80,731 healthy controls, **Table 1**). Genotypes for both variants were extracted from these datasets and tested for association by meta-analysis together with the discovery cohort (see Methods). Meta-analysis of all cohorts together increased the evidence for association of both variants with AIS risk (**Figure 1B**). While a similar effect size was

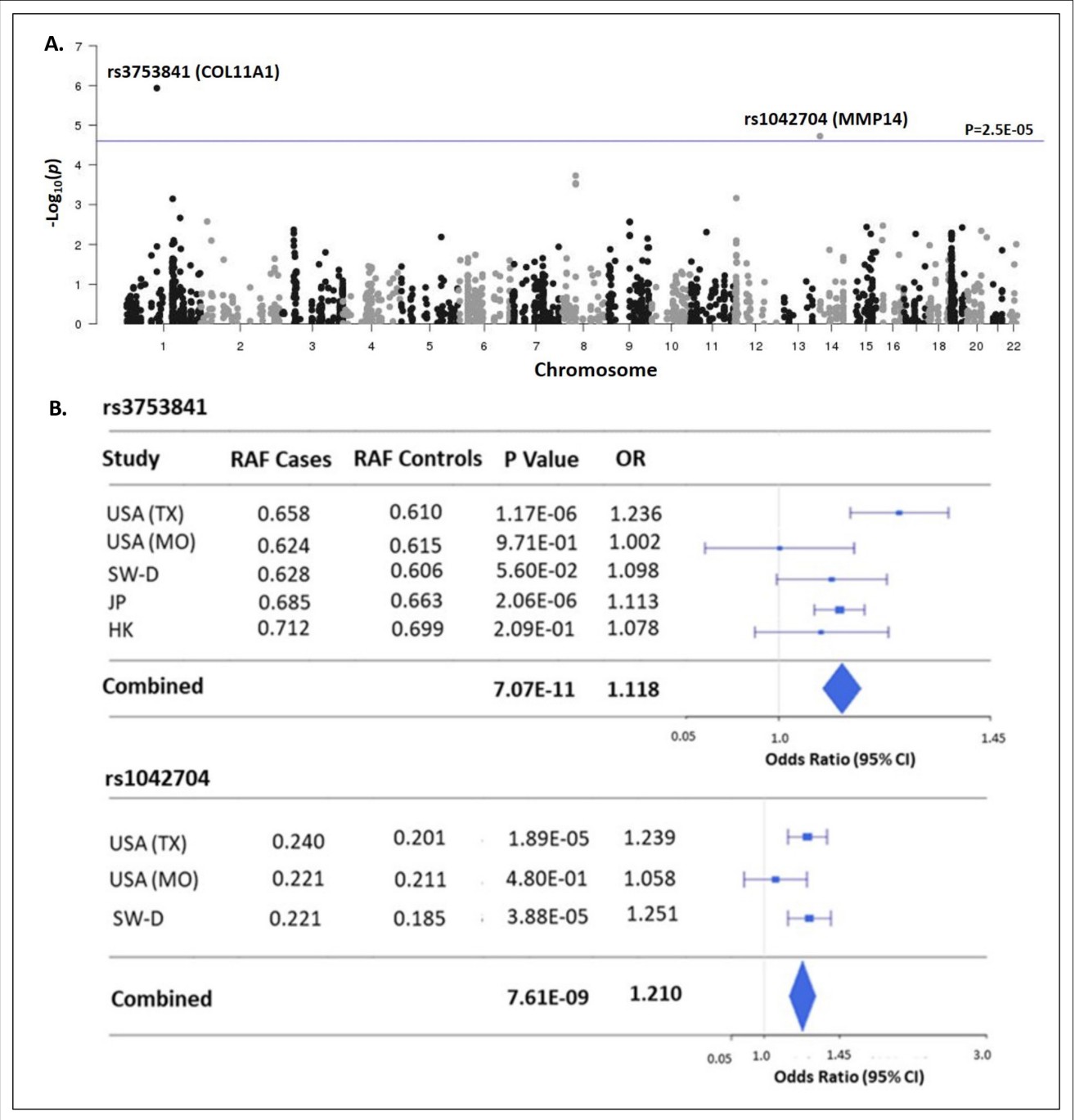

**Figure 1.** Matrisome-wide association study. (**A**) Manhattan plot showing –log10 p-values (y-axis) versus chromosomal position (x-axis) for the 2008 common coding variants tested in the discovery study USA (TX). The horizontal line represents the threshold for significance level (p-value <2.5 × 10⁻⁵) after Bonferroni multiple testing correction. (**B**) Tests of association for SNPs rs3753841 and rs1042704 in discovery and independent replication cohorts. RAF – reference allele frequency; OR – odds ratio; CI –confidence interval.

The online version of this article includes the following figure supplement(s) for figure 1:

**Figure supplement 1.** Statistical power as a function of the genotype relative risk (OR) to detect significant association at α=2.5E⁻⁰⁵ for different disease allele frequencies, using 1358 cases and 12,507 controls in the discovery study.

**Figure supplement 2.** Manhattan plot showing –log10 p-values (y-axis) plotted versus chromosomal position (x-axis) for the 2009 common coding variants tested for females in the discovery study USA (TX).

**Figure supplement 3.** Tests of association of SNP rs1042704 with adolescent idiopathic scoliosis (AIS) in East Asian cohorts.

**Figure supplement 4.** LocusZoom plots of SNPs in genomic regions of SNPs rs3753841 (top) and rs1042704 (bottom).

noted for rs1042704 in Japanese and Han Chinese cohorts, the results were less significant, likely due to lower minor allele frequencies (East Asian MAF = 0.02 compared to total non-Asian cohort MAF = 0.20) in these populations (*Figure 1—figure supplement 3*). Plotting recombination across both regions suggested that these signals were likely confined to blocks of linkage disequilibrium within the *COL11A1* and *MMP14* genes, respectively (*Figure 1—figure supplement 4*).

Rare dominant mutations in *COL11A1,* often disrupting a Gly-X-Y sequence, can cause Marshall (MRSHS) (OMIM #154780) or Stickler syndromes (STL2) (OMIM #604841) marked variously by facial anomalies, sensineural hearing loss, short stature, spondyloepiphyseal dysplasia, eye anomalies, ectodermal features, and scoliosis. Notably, our AIS cohort and particularly individuals carrying the rs3753841 risk allele were negative for co-morbidities or obvious features of Marshall or Stickler syndromes. Thus, variation in *COL11A1* is independently associated with AIS. Notably, we did not detect common variants in linkage disequilibrium ($R^2>0.6$) with the top SNP rs3753841 (*Figure 1—figure supplement 4*). Further, analysis of 625 exomes from the discovery cohort (46%) identified only three rare *COL11A1* variants in five individuals (*Supplementary file 2*), and rare variant burden testing for *COL11A1* was not significant as expected (data not shown). These observations suggested that rs3753841 itself could confer disease risk, although our methods would not detect deep intronic variants that could contribute to the overall association signal.

## COL11A1 is expressed in adolescent spinal tissues

We next characterized *COL11A1* in postnatal spine development. *COL11A1* encodes one of three alpha chains of type XI collagen, a member of the fibrillar collagen subgroup and regulator of nucleation and initial fibril assembly, particularly in cartilage (*Fernandes et al., 2007*). Spinal deformity is well described in *Col11a1*-deficient (*cho/cho*) embryos (*Hafez et al., 2015*; *Seegmiller et al., 1971*). In mouse tendon, *Col11a1* mRNA is abundant during development but barely detectable at 3 months of age (*Wenstrup et al., 2011*). We analyzed RNAseq datasets derived from adolescent human spinal tissues (*Makki et al., 2020*), finding that *COL11A1* was upregulated in cartilage relative to bone and muscle. In cartilage, *PAX1* and *COL11A2* showed the strongest expression levels relative to other published human AIS-associated genes (*Kou et al., 2013*; *Ogura et al., 2015*; *Sharma et al., 2015*; *Khanshour et al., 2018*; *Haller et al., 2016*; *Baschal et al., 2014*; *Gao et al., 2007*; *Figure 2A*). In all, most AIS-associated genes showed the strongest expression levels in cartilage relative to other adolescent spinal tissues.

We next sought to characterize *Col11a1* expression in spines of postnatal mice. To detect COL11A1 protein (collagen α1(XI)), we performed immunohistochemistry (IHC) and immunofluorescence (IF) microscopy using a collagen α1(XI) reactive antibody (*Sun et al., 2020*) in newborn (P0.5) and adolescent (P28) mice. In spines of P0.5 mice, strong staining was observed in the nucleus pulposus (NP) and in surrounding annulus fibrosus (AF) (*Figure 2B*). In thoracic spines of P28 mice, the compartments of the IVD were more distinct, and strong collagen α1(XI) staining was observed in each (*Figure 2C*). In regions of the cartilage endplate (CEP)-vertebral bone transition, collagen α1(XI) was detected in columnar chondrocytes, particularly in the hypertrophic zone adjacent to condensing bone (*Figure 2C*). We also examined collagen α1(XI) expression in ribs, as these structures are also involved in the scoliotic deformity (*Richards et al., 2020*). In P28 rib growth plates, as in spine, a biphasic pattern was observed in which collagen α1(XI) reactivity was most pronounced around cells of the presumed resting and pre-hypertrophic/hypertrophic zones (*Figure 2—figure supplement 1*). These data show that in mouse, collagen α1(XI) is detectable in all compartments of young postnatal IVD and, at the thoracic level, is particularly abundant in the chondro-osseous junction region of IVD and vertebral growth plate.

## Col11a1 is downregulated in the absence of Pax1 in mouse spine and tail

We previously identified AIS-associated variants within a putative enhancer of *PAX1* encoding the transcription factor Paired Box 1 (*Sharma et al., 2015*; *Khanshour et al., 2018*). Pax1 is a well-described marker of condensing sclerotomal cells as they form segments that will eventually become the IVD and vertebrae of the spine (*Chan et al., 2014*; *Aszódi et al., 1998*; *Smith et al., 2011*). We generated *Pax1* knockout mice (*Pax1$^{-/-}$*) using CRISPR-Cas9 mutagenesis and validated them using sequencing and southern blot (*Figure 3—figure supplement 1*). Homozygous *Pax1$^{-/-}$* mice were viable

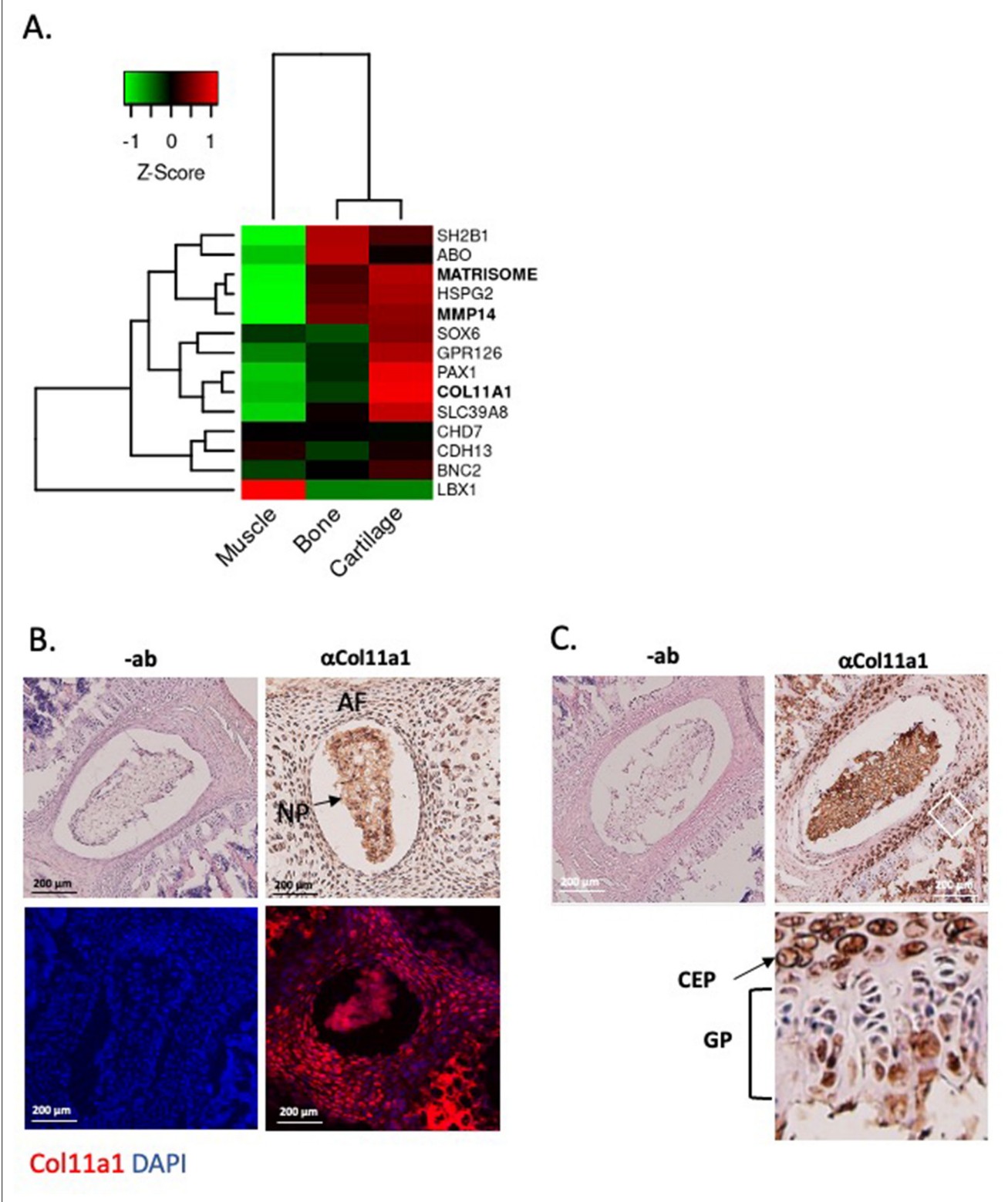

**Figure 2.** *Col11a1* and *Mmp14* expression in spine. (**A**) A heatmap of transcript per million (TPM) values of *COL11A1*, *MMP14*, and other published genes associated with adolescent idiopathic scoliosis (AIS). The average TPM value of matrisome genes is represented as MATRISOME. (**B**) Detection of collagen a1(XI) in P0.5 mouse spine. Immunohistochemistry (IHC) shown at top, with immunofluorescence (IF) staining below. '-ab' refers to negative controls lacking primary antibody (shown at left). Results are representative of N≥3 technical replicates in whole spines. (**C**) Detection of collagen a1(XI)

*Figure 2 continued on next page*

*Figure 2 continued*

in P28 mouse spine. Negative antibody IHC control shown at left; antibody-positive IHC shown at right. Enlarged, rotated view of white boxed area shows a biphasic staining pattern. CEP – cartilage endplate; GP – growth plate. Results are representative of N≥3 technical replicates in whole spines.

The online version of this article includes the following figure supplement(s) for figure 2:

**Figure supplement 1.** Immunofluorescence (IF) staining using collagen a1(XI) antibody in P28 ribs (top).

and developed spine deformity and kinks in the tail, as observed in other *Pax1*-deficient mice (***Wilm et al., 1998***). We next compared the expression of collagen α1(XI) protein in IVD and condensing bone of wild-type and *Pax1*-/- mice by performing IF staining in P28 spines (***Figure 3A***). In wild-type IVD, strong overlapping expression of collagen α1(XI) and PAX1 cells was observed, mostly within the CEP and chondro-osseous interface (***Figure 3A***). PAX1 staining was negative in *Pax1*-/- mice as expected, and collagen α1(XI) staining was dramatically diminished in CEP and the chondro-osseous vertebral borders. Moreover, the IVD in *Pax1*-/- mice was highly disorganized, without discernable NP, AF, and CEP structures as has been reported (***Figure 3—figure supplement 2***; ***Wallin et al., 1994***). To test the effect of *Pax1* on expression of *Col11a1* and other AIS-associated genes during embryonic development, RNA was isolated from vertebral tissue dissected from the tails of embryonic stage 12.5 (E12.5) wild-type and *Pax1*-/- mice and subjected to bulk RNAseq and quantitative real-time PCR (qRT-PCR) (***Figure 3B***). Gene-set enrichment analysis of RNAseq was most significant for the gene ontology term 'extracellular matrix' (***Figure 3C***). By qRT-PCR analysis, expression of *Col11a1*, *Adgrg6*, and *Sox6* was significantly reduced in female and male *Pax1*-/- mice compared to wild-type mice (***Figure 3D–G***). These data show that loss of *Pax1* leads to reduced expression of *Col11a1* and the AIS-associated genes *Adgrg6* and *Sox6* in affected tissue of the developing tail.

## Col11a1 regulates Mmp3 expression in chondrocytes

*COL11A1* has been linked with ECM remodeling and invasiveness in some cancers (***Wu et al., 2014***). In solid tumors, *COL11A1* has been shown to alter ECM remodeling by enhancing *MMP3* expression in response to TGFꞵ1 (***Wu et al., 2014***). *MMP3* encodes matrix metalloproteinase 3, also known as stromolysin, an enzyme implicated in matrix degradation and remodeling in connective tissues (***Mudgett et al., 1998***). We confirmed strong *MMP3* mRNA expression, relative to *COL11A1*, in human spinal cartilage and bone, but minimal expression in spinal muscle (***Figure 4—figure supplement 1***). We next cultured costal chondrocytes from P0.5 *Col11a1*<sup>fl/fl</sup> mice (***Sun et al., 2020***) and subsequently removed *Col11a1* by treating with Cre-expressing adenoviruses. After confirming *Col11a1* excision (***Figure 4A***), we compared *Mmp3* expression in these cells to cells treated with GFP-expressing adenoviruses lacking Cre activity. We found that *Mmp3* expression was significantly increased in cells where *Col11a1* mRNA expression was downregulated by about 70% compared to untreated cells (***Figure 4B***). Furthermore, western blotting in these cells demonstrated an ~2- to 5-fold increase in pro-, secreted, and active forms of Mmp3 protein when collagen α1(XI) was reduced. The proteolytic processing per se of precursor MMP3 into active forms (***Sun et al., 2014***) did not appear to be affected by *Col11a1* expression (***Figure 4C***). These results suggest that *Mmp3* expression is negatively regulated by *Col11a1* in mouse costal chondrocytes.

To test whether *Col11a1* affects *Mmp3* expression in vivo, we bred *Col11a1*<sup>fl/fl</sup> female mice with *Col11a1*<sup>fl/fl</sup>:*ATC* males carrying the Acan enhancer-driven, doxycycline-inducible Cre (ATC) transgene (***Dy et al., 2012***). ATC has been shown to harbor Cre-mediated recombination activity in most differentiated chondrocytes and in NP within 2 days of treating pregnant mothers with doxycycline starting at E15.5 (***Dy et al., 2012***). ATC activity was confirmed by crossing this line to the R26td<sup>[Tomato]</sup> reporter that ubiquitously expresses the fluorescent gene Tomato after Cre recombination. Strong Cre activity was seen in P0 pups of mothers treated with doxycycline at E15.5 in the NP, CEP, and AF of the IVD and in chondrocytes of the growth plates (***Figure 4—figure supplement 2***). Pregnant *Col11a1*<sup>fl/fl</sup> females were treated with doxycycline water from E15.5 to induce Cre expression in differentiated chondrocytes. Excision of *Col11a1* was confirmed in DNA from costal cartilage of *Col11a1*<sup>fl/fl</sup>:*ATC*-positive offspring (***Figure 4—figure supplement 3***). Consistent with results obtained by in vitro excision of *Col11a1*, cartilage from mice deficient in *Col11a1* showed ~4-fold upregulation of *Mmp3* mRNA expression relative to *Col11a1*<sup>fl/fl</sup> mice (***Figure 4D***).

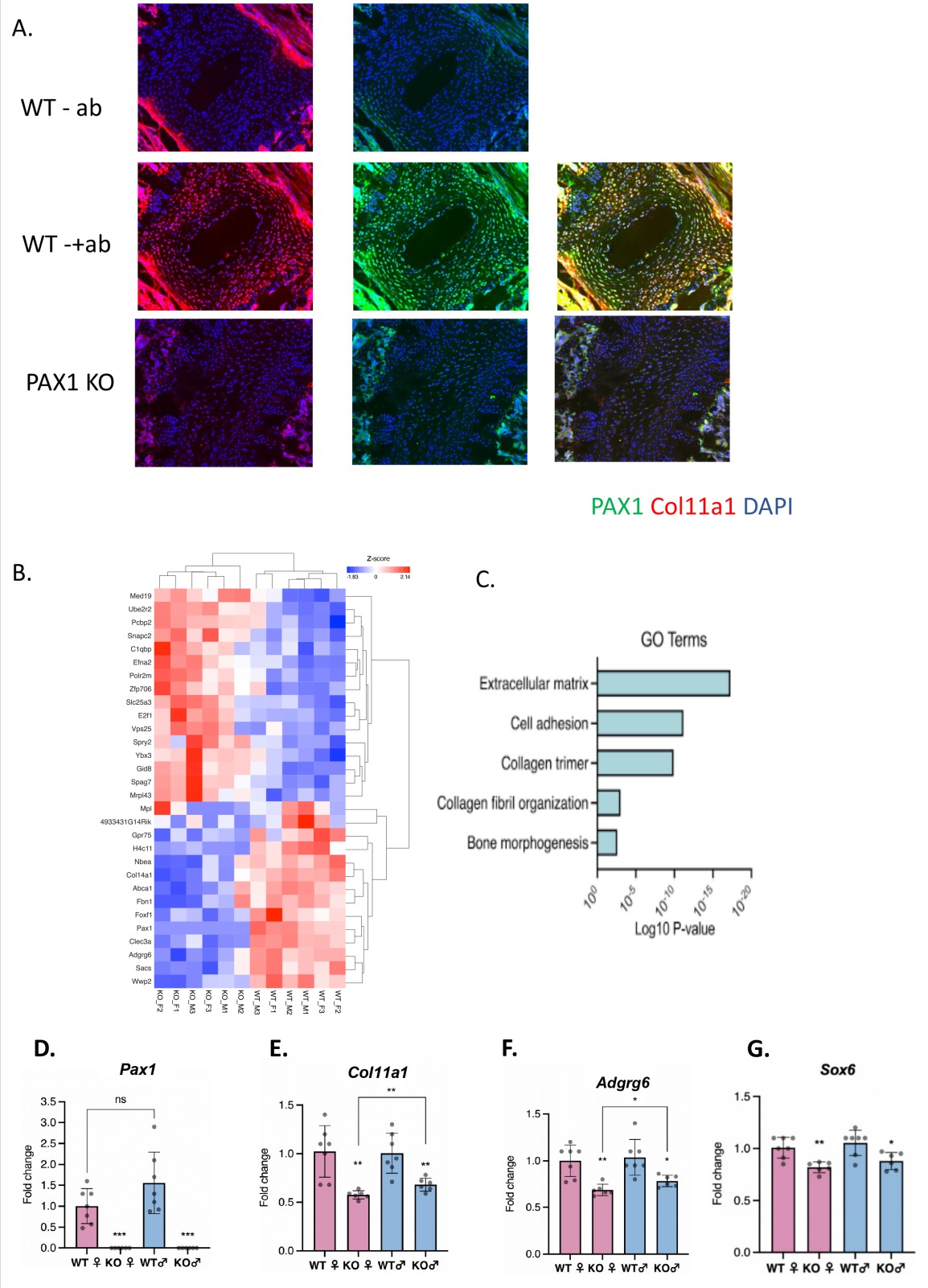

**Figure 3.** Assessing *Pax1* regulation of *Col11a1 expression*. (**A**) Immunofluorescence (IF) staining of P28 intervertebral disc (IVD) from thoracic regions of *Pax1$^{-/-}$* (bottom) and wild-type (WT) littermate (middle, top) mice using PAX1- (green) and collagen a1(XI)-specific (red) antibodies and DAPI nuclear counterstain. Antibody-negative controls are shown at top as (-ab). Results are representative of N≥3 technical replicates in whole spines. (**B**) Heatmap of differentially expressed genes (p-value <0.0001) in embryonic stage 12.5 (E12.5) tails of WT and *Pax1$^{-/-}$* mice. (**C**) Gene ontology (GO) analysis of

*Figure 3 continued on next page*

Figure 3 continued

differentially expressed genes in E12.5 tail WT and *Pax1*[-/-] mice. (**D–G**) Gene expression levels dissected from E12.5 mouse tail from WT and *Pax1*[-/-] (knockout [KO]) mice as determined by quantitative real-time PCR (qRT-PCR). Each value represents the ratio of each gene expression to that of β-actin, and values are mean ± standard deviation. The expression value of WT female group was arbitrarily set at 1.0. Each dot represents one embryo and statistical differences were determined using a two-sided unpaired t-test (*p<0.05, **p<0.01, ***p<0.001).

The online version of this article includes the following figure supplement(s) for figure 3:

**Figure supplement 1.** Design and validation of Pax1 knockout in mouse using CRISPR-mediated gene targeting.

**Figure supplement 2.** HE staining of sectioned lumbar spines from wild-type (left) and *Pax1*[-/-] (right) mice.

## AIS-associated variant in COL11A1 perturbs its regulation of MMP3

Although low-resolution structures currently available for collagen triple helices are not useful for modeling the effects of individual variants on protein stability, we noted that the AIS-associated variant P1335L occurs at the third position of a Gly-X-Y repeat and consequently could be structurally important in promoting stability of the triple helix, particularly if it is hydroxylated. We also noted that this variant is predicted to be deleterious by Combined Annotation Dependent Depletion (CADD) (*Rentzsch et al., 2021*) and Genomic Evolutionary Rate Profiling (GERP) (*Cooper et al., 2005*) analysis (CADD = 25.7; GERP = 5.75). Further, *COL11A1* missense variants have been shown to evoke transcriptional changes in ECM genes in cancer cells (*Lee et al., 2021*). We therefore tested whether the *COL11A1*[P1335L] sequence variant alters its regulation of *Mmp3* in chondrocytes. For this, SV40-immortalized cell lines were established from *Col11a1*[fl/fl] mouse costal chondrocytes and transduced with lentiviral vectors expressing green fluorescent protein (GFP) and *COL11A1*[wt], *COL11A1*[P1335L], or vector alone. After transduction, GFP-positive cells were grown to 50% confluence and treated with Cre-expressing adenovirus (ad5-Cre) to remove endogenous mouse *Col11a1* (*Figure 5A*). Using a human-specific *COL11A1* qRT-PCR assay, we detected overexpression of *COL11A1*[wt] and *COL11A1*[P1335L] compared to untransduced cells regardless of Cre expression (*Figure 5A*). Western blotting with an antibody directed against the HA epitope tag confirmed overexpression of human collagen α1(XI) protein (*Figure 5B*). Endogenous *Mmp3* mRNA and protein upregulation was evident by qRT-PCR and western blotting, respectively, in untransduced cells treated with Ad5-Cre, as expected. Overexpressing human wild-type *COL11A1* suppressed *Mmp3* expression, consistent with the negative regulation we previously observed (*Figure 5A and B*). However, the *COL11A1*[P1335L] mutant failed to downregulate *Mmp3* expression despite being overexpressed (*Figure 5A and B*). Thus, regulation of *Mmp3* appeared to be perturbed in the presence of the *COL11A1*[P1335L] variant in these cells.

## *Col11a1* and *Mmp3* are responsive to estrogen receptor signaling in chondrocytes

The expression of *Col11a1*, and of other ECM genes, is known to be estrogen-responsive in certain tissues, such as ovarian follicular cells (*Zalewski et al., 2012*). Because of the suspected role of endocrine hormones in AIS, we investigated whether *Col11a1* expression was responsive to estrogen receptor siRNA-mediated knockdown in cultured chondrocytes. We first validated that *Mmp3* mRNA and protein levels were significantly increased after *Col11a1* knockdown in wild-type chondrocytes, as observed by Cre-mediated deletion in *Col11a1*[fl/fl] chondrocytes (*Figure 6A*). Estrogen receptor 2 (*Esr2*), but not estrogen receptor alpha (*Esr1*), was detected in mouse chondrocytes by qRT-PCR (data not shown). We therefore tested the consequences of *Esr2* siRNA-mediated knockdown on gene expression in chondrocytes. After *Esr2* knockdown, *Col11a1* as well as *Pax1* was significantly upregulated compared to scramble control, while *Mmp3* expression was significantly downregulated (*Figure 6B*). We also performed *Col11a1* knockdowns in these cells and noted upregulation of *Pax1* expression, suggesting a negative feedback loop between *Pax1* and *Col11a1* in these cells (*Figure 6B*). Simultaneous knockdown of *Col11a1* and *Esr2* expression reduced *Mmp3* expression to normal levels, supporting a possible interaction between *Col11a1* and *Esr2* in regulating *Mmp3*. Treating chondrocytes with tamoxifen, an estrogen receptor modulator, also upregulated *Col11a1* expression to similar levels as observed after *Esr2* knockdown, compared to cells treated with DMSO carrier (*Figure 6— figure supplement 1*). These results suggest that estrogen signaling suppresses *Col11a1* expression. In cultured rat CEP cells, *Esr2* mRNA was downregulated, and *Mmp3* mRNA was upregulated after

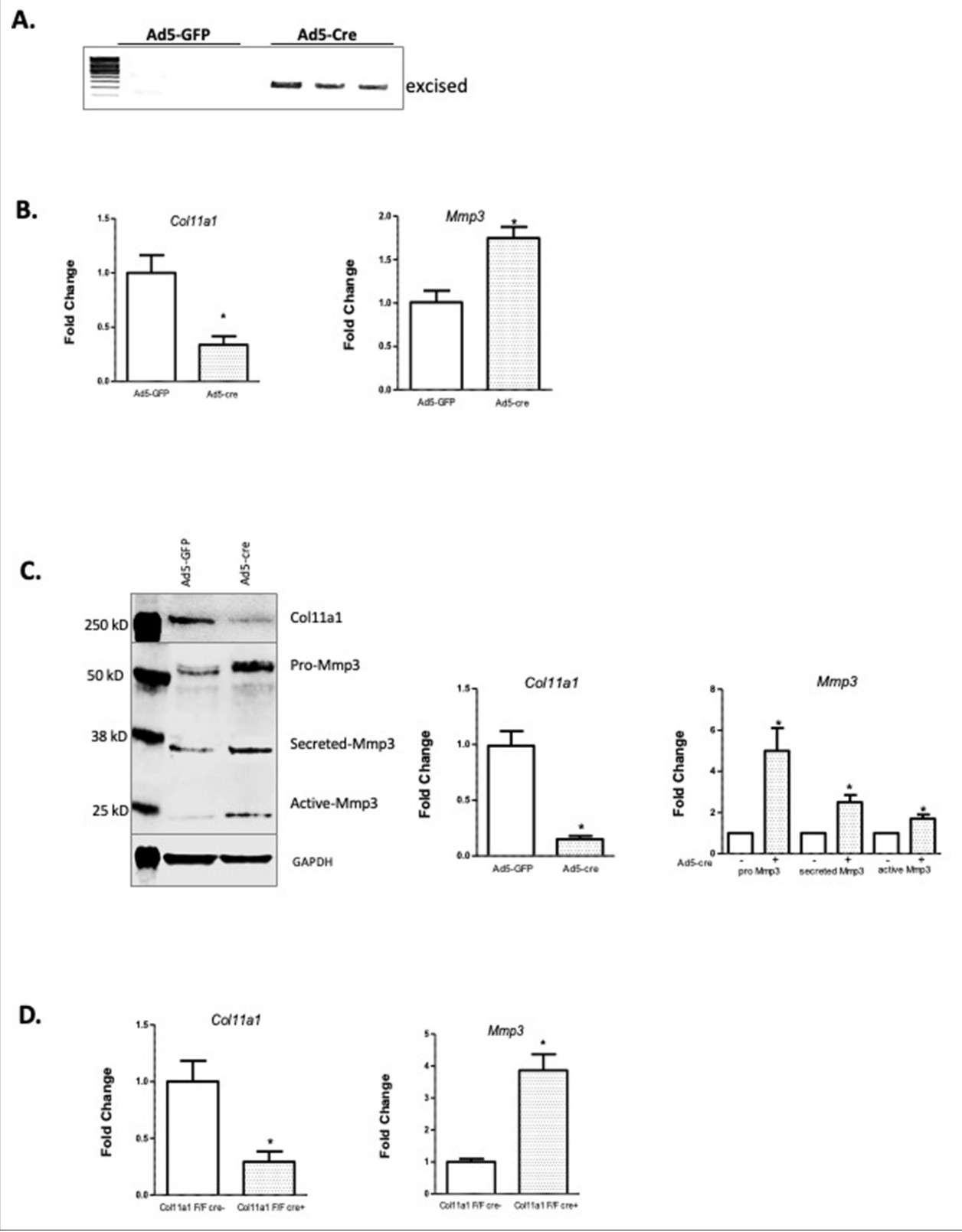

**Figure 4.** *Col11a1* regulation of *Mmp3* expression in cartilage. (**A**) PCR assay of *Col11a1* excision in *Col11a1*^fl/fl^ cultured costal chondrocytes. (**B**) Gene expression levels from *Col11a1*^fl/fl^ cultured costal chondrocytes transduced with green fluorescent protein (GFP) (Ad5-GFP, left) or Cre-expressing adenovirus (Ad5-cre, right) as determined by quantitative real-time PCR (qRT-PCR). Values represent the ratio of each gene expression to that of *GAPDH*, and values are mean ± standard deviation. The expression value of control Ad5-GFP results was arbitrarily set at 1.0. Statistical differences

*Figure 4 continued on next page*

*Figure 4 continued*

were determined using a two-sided paired t-test (*p<0.05). Results shown for N≥3 biologic replicates, each including three technical replicates. (**C**) Western blot detection of collagen a1(XI), MMP3, and GAPDH loading control in cultured costal chondrocytes after Ad5-GFP or Ad5-cre transduction. Results are representative of N=4 biologic replicates. Protein size ladder is shown in lane 1. Quantification of bands detected by western blotting, where Ad5-GFP was set to 1.0, is shown at right. Statistical differences were determined using a two-sided paired t-test (*p<0.05). (**D**) Gene expression levels from dissected *Col11a1^fl/fl^:ATC* costal cartilage, analyzed as described in (**A**). Results shown for N=3 biologic replicates, each including three technical replicates.

The online version of this article includes the following source data and figure supplement(s) for figure 4:

**Source data 1.** Original gel images of *Col11a1 ^fl/fl^* excision PCR assay in *Figure 4A*.

**Source data 2.** *Figure 4A* and original gel images of *Col11a1 ^fl/fl^* excision PCR assay with highlighted and labeled bands.

**Source data 3.** Original western blot images (anti-COL11A1, anti-MMP3, anti-GAPDH) shown in *Figure 4C*.

**Source data 4.** *Figure 4C* and original western blot images (anti-COL11A1, anti-MMP3, anti-GAPDH) with highlighted bands and labels.

**Figure supplement 1.** Relative expression of *MMP3* compared to *COL11A1* in human spinal tissues.

**Figure supplement 2.** Immunofluorescence microscopy of Rosa26^+/-^:ATC P0 spines, without doxycycline treatment (left) and after doxycycline treatment starting at embryonic stage 15.5 (E15.5) (right).

**Figure supplement 3.** PCR assays in DNA from costal cartilage.

**Figure supplement 3—source data 1.** Original gel images of *Col11a1 ^fl/fl^* excision PCR assay in *Figure 4—figure supplement 3*.

**Figure supplement 3—source data 2.** Original gel images of *Col11a1 ^fl/fl^* excision PCR assay in *Figure 4—figure supplement 3* with highlighted and labeled bands.

*Col11a1* knockdown, as observed in mouse chondrocytes (*Figure 6C*, *Figure 6—figure supplement 2*). However, *Esr2* knockdown did not significantly impact *Col11a1* or *Mmp3* expression in these cells (*Figure 6C*). Hence, we conclude that in cultured mouse chondrocytes, ESR2 signaling disrupts the suppression of *Mmp3* by *Col11a1*.

## Discussion

AIS has been described in the medical literature for centuries, yet its underlying etiology has remained enigmatic (*Wise and Sharma, 2010*). Given that AIS originates in children who appear to be otherwise healthy, even its tissue of origin has been difficult to discern, and long debated (*Wise et al., 2020*). The advent of powerful genotyping and sequencing methods in the last two decades has led to breakthrough discoveries of genetic loci associated with AIS, most in non-coding regions of the genome that are difficult to interpret biologically (*Wise et al., 2020*). Aggregating these results, however, provided supportive evidence that pathways of cartilage and connective tissue ECM development are relevant in AIS etiology (*Wise et al., 2020*; *Khanshour et al., 2018*). Here, in the largest multi-ethnic human cohort studied to date, we elected to test the hypothesis that alterations in ECM proteins themselves contribute to AIS susceptibility. This approach yielded most significant evidence for a common protein-altering variant in the *COL11A1* gene encoding collagen α1(XI), a minor yet critical component of cartilaginous ECM. Moreover, our studies define a *COL11A1*-mediated disease pathway (*Figure 7*) and point to the chondro-osseous junction of IVD and vertebrae spine as a relevant cellular compartment in AIS etiology.

The results of this study together with the previous observation of *COL11A2* rare variant enrichment in AIS support a role for the collagen α1(XI) heterotrimer itself in its pathogenesis (*Haller et al., 2016*). Collagen type XI, composed of three chains encoded by the *COL11A1*, *COL11A2*, and *COL2A1* genes (OMIM #s 120280,120290, 120140, respectively), is a minor component of collagen type II fibrils that are abundant in cartilage. Collagen type XI is also broadly expressed in testis, trachea, tendons, trabecular bone, skeletal muscle, placenta, lung, brain neuroepithelium, the vitreous of the eye, and IVDs (*Yoshioka et al., 1995*). In the pericellular space, collagen α1(XI) initiates fibrillogenesis with collagen type II fibrils, maintaining regular spacing and diameter of the collagen fibrils, while organizing the pericellular surface by interaction with cartilage proteoglycans (*Smith et al., 1989*; *Luo and Karsdal, 2019*). Purified human collagen type XI, when added back to chondrocytes in *in vitro* culture, stimulates chondrogenesis while inhibiting hypertrophy, as measured by histological staining, proliferation assays, and relative expression of chondrogenic early marker genes (*Li et al., 2018*). In newborn and 1-month-old mice, we found that collagen

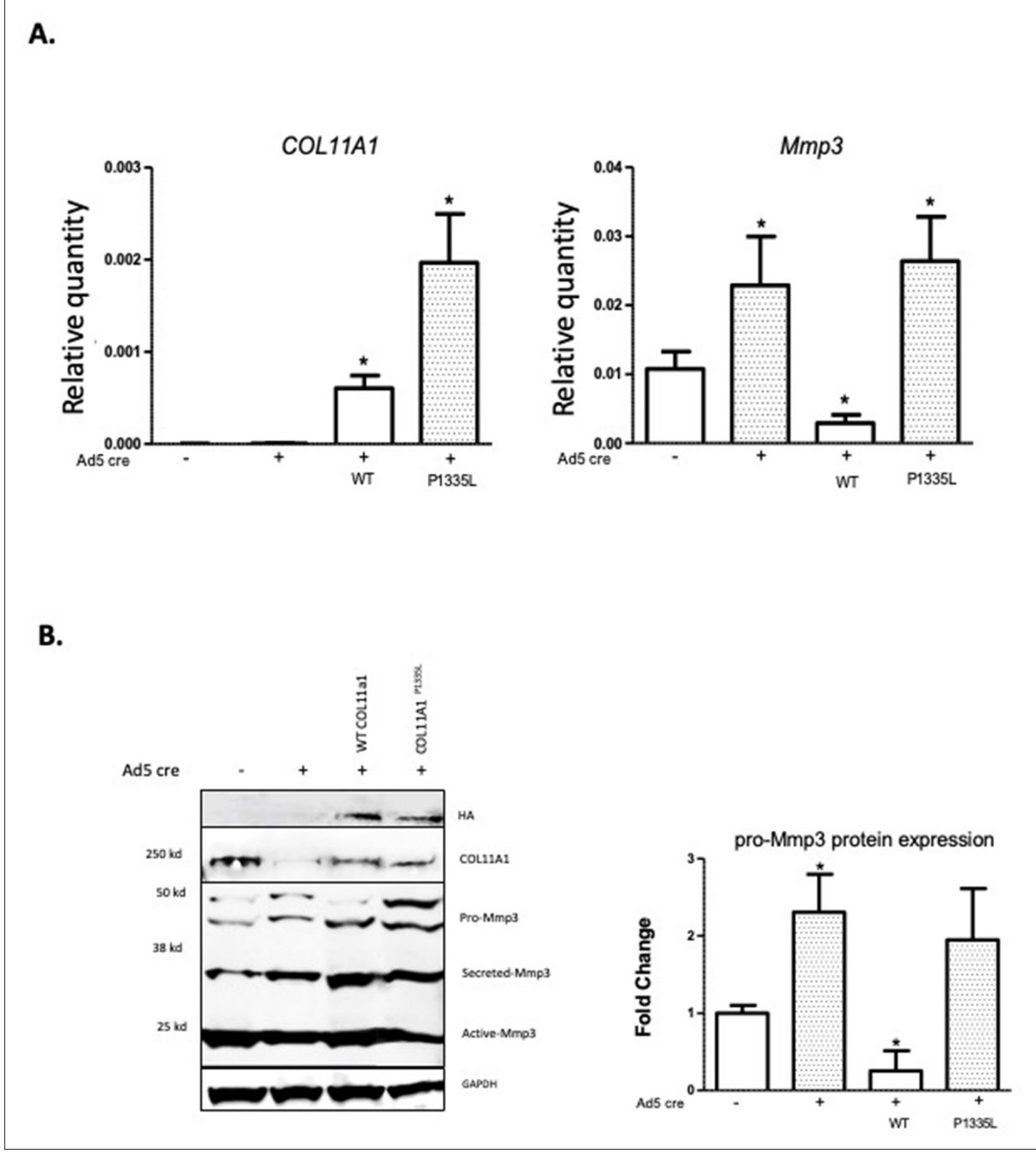

**Figure 5.** *Col11a1^P1335L* regulation of *Mmp3* expression in lentiviral transduced mouse GPCs. (**A**) Quantitative real-time PCR (qRT-PCR) of human *COL11A1* and endogenous mouse *Mmp3* in SV40-immortalized mouse costal chondrocytes transduced with the lentiviral vector only (lanes 1,2), human wild-type (WT) *COL11A1* (lane 3), or *COL11A1^P1335L^*. Values represent the ratio of each gene expression to that of *GAPDH*, and values are mean ± standard deviation. Significant quantitative changes (p≤0.05) relative to vector-only transfected cells as measured by unpaired t-tests are shown by *. Results shown for N=4 biologic replicates, each including three technical replicates. (**B**) Western blot corresponding to experiments shown in (**A**) using HA antibody to detect epitope-tagged human collagen a1(XI), COL11A1 antibody to detect mouse and human collagen a1(XI), MMP3 antibody to detect endogenous mouse MMP3, and GAPDH. Values are mean after normalization to GAPDH, ± standard deviation. Significant differences (p≤0.05) relative to vector-only, Ad5-negative transfected cells as measured by unpaired t-tests are shown by *.

The online version of this article includes the following source data for figure 5:

**Source data 1.** Original western blot images (anti-COL11A1, anti-MMP3, anti-GAPDH) with highlighted bands and labels.

**Source data 2.** *Figure 5B* and original western blot images (anti-COL11A1, anti-MMP3, anti-GAPDH) with highlighted bands and labels.

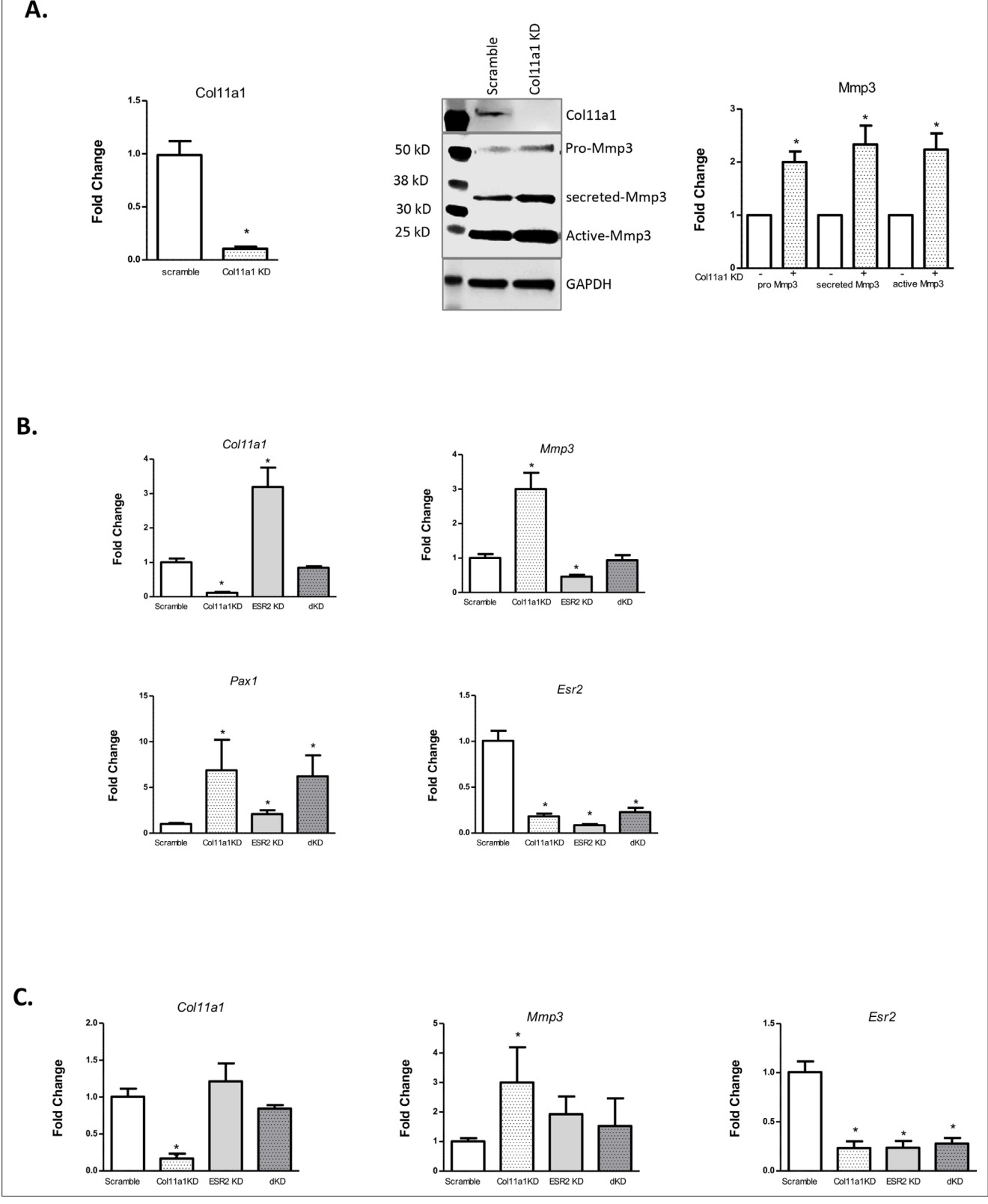

**Figure 6.** Effects of estrogen receptor beta on *Col11a1-Mmp3* signaling axis. (**A**) RT-qPCR (left) of *Col11a1* expression after siRNA-mediated knockdown as shown at left. Representative western blot (of N=4 biologic replicates) of cultured costal chondrocytes after scramble or *Col11a1*-specific siRNA knockdown is shown in middle. Protein size ladder is shown in lane 1. Quantification of bands detected by western blotting is shown at right, where scramble results were set to 1.0. Values are mean after normalization to GAPDH, ± standard deviation. (**B**) Gene expression levels of *Col11a1*, *Mmp3*,

*Figure 6 continued on next page*

*Figure 6 continued*

*Pax1*, and *Esr2* mRNA in cultured costal chondrocytes showing fold change relative to the scramble control. dKD = double *Col11a1-Esr2*-specific siRNA knockdowns. Each value represents the ratio of each gene expression to that of *GAPDH*, and values are mean ± standard deviation. Results are representative of N≥3 biologic replicates, each including three technical replicates. (**C**) Gene expression levels from rat cartilage endplate (CEP) cells, as described in (**B**).

The online version of this article includes the following source data and figure supplement(s) for figure 6:

**Source data 1.** Original western blot images (anti-COL11A1, anti-MMP3, anti-GAPDH) with highlighted bands and labels.

**Source data 2.** *Figure 6A* and original western blot images (anti-COL11A1, anti-MMP3, anti-GAPDH) with highlighted bands and labels.

**Figure supplement 1.** Quantitative real-time PCR (qRT-PCR) of *Col11a1* and *Col11a2* mRNA in cultured costal chondrocytes treated with DMSO carrier or tamoxifen (N≥3 independent experiments).

**Figure supplement 2.** Quantitative real-time PCR (qRT-PCR) of *Sfrp2*, *Krt19,* and *Mmp12* mRNA to validate expression of these marker genes in cultured rat nucleus pulposus (NP), annulus fibrosus (AF), and cartilage endplate (CEP) cells.

α1(XI) was abundant in IVD and at the chondro-osseous junction of IVD and vertebrae, particularly concentrated in pre-hypertrophic/hypertrophic chondrocytic cells. In long bone growth plates, *Long et al., 2022*, recently identified eight distinct cell clusters after unsupervised analysis of single cell (scRNAseq) of flow-sorted hypertrophic chondrocytes from Col10a1Cre;Rosa26fs-tdTomato mice. At E16.5, *Col11a1* expression was highest in cells with signatures of pre-hypertrophic to hypertrophic transition, and lowest in cells with osteogenic signatures (M Hilton, personal communication) (*Long et al., 2022*). Taken together, these results suggest that collagen α1(XI) normally participates in maintaining growth plate cells in a hypertrophic, pre-osteogenic state, although little is known about its precise molecular function in that compartment, or in the IVD, during spinal development. Spines of *Col11a1*-deficient mice (*cho/cho*) show incompletely formed vertebral

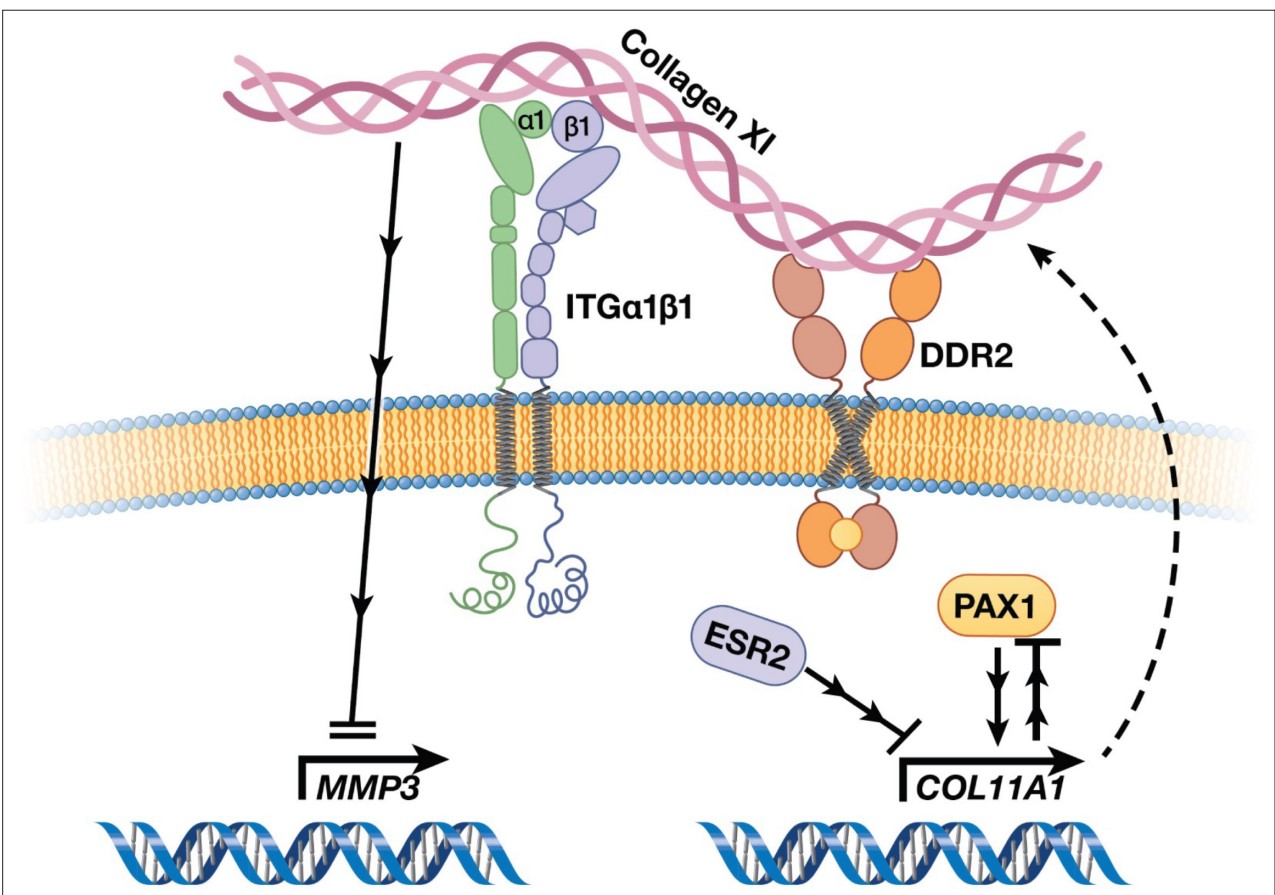

**Figure 7.** Cartoon depiction of a collagen XI-mediated signaling axis in chondrocytes. Collagen XI is held in the pericellular space by integrins and DDR2. COL11A1, under the regulation of ESR2 and PAX1, signals through unknown mechanisms and inhibits MMP3 transcription.

bodies, spinal curvatures, and decreased separation between vertebrae, which are themselves less mineralized than in wild-type mice (*Hafez et al., 2015*). Notably, common *COL11A1* variants also have been associated with adult lumbar disc herniation and lumbar disc degeneration, as well as DXA-measured bone size, spinal stenosis, and spondylolisthesis (*Jiang et al., 2017*; *Mio et al., 2007*; *Styrkarsdottir et al., 2019*). Although gain-of-function or dominant-negative effects of the rs3753841 variant would not have been revealed in our assays, the spinal deformity noted in the *cho/cho* loss-of-function model, and failure of missense variants in *Col11a2* to rescue congenital scoliosis (*Rebello et al., 2023*), leads us to surmise that reduction in the components of collagen type XI disrupts spinal development.

*Pax1* is a well-described marker of early spine development, where it activates a gene expression cascade starting at E12.5–13.5 in mouse development (*Wilm et al., 1998*; *Rodrigo et al., 2003*; *Sivakamasundari et al., 2017*). Our data showed that loss of *Pax1* leads to decreased expression of *Col11a1*, *Sox6*, and *Adgrg6* in E12.5 tails of both male and female mice. The downregulation of *Col11a1* is consistent with a prior study of gene expression in flow-sorted GFP-labeled *Pax1⁻/⁻* embryonic IVD cells (*Sivakamasundari et al., 2017*). However, from these experiments we cannot discern if *Pax1* directly regulates *Col11a1* in cis, or by an indirect effect. It is likely, however, that *Col11a1* expression in developing tail is directly activated by binding SOX transcription factors, as a prior genomic study using chromatin immunoprecipitation and sequencing in rat chondrosarcoma cells identified super enhancers near the *Col11a1* gene that were bound multiple times by SOX9 and SOX6 (*Liu and Lefebvre, 2015*). The SOX5/6/9 trio is known to regulate many of the same genes as PAX1 (*Sivakamasundari et al., 2017*), but whether this includes *Col11a1* is unknown.

In mouse postnatal spines, we observed co-localization of collagen α1(XI) and PAX1 proteins specifically within the cartilaginous endplate-vertebral junction region that includes the vertebral growth plate. The endplate, which is important as the site allowing diffusion of nutrients from the circulation into the avascular inner IVD, harbors subpopulations of cells expressing type II collagen presumably organized by collagen type XI (*Chan et al., 2014*; *Smith et al., 2011*). While the endplate is continuous with the vertebral growth plate in mice, it is important to note that in humans the endplate and vertebrae become distinctly separate structures with closure of the growth plates at puberty (*Chan et al., 2014*). This is also the site of the ring apophyses that form the insertion of the IVD into vertebrae (*Costa et al., 2021*). Lagging maturity of the ring apophysis, combined with mechanical forces across the IVD in the horizontal plane, has been proposed as an initiating factor leading to rotatory decompensation in the adolescent spine in AIS (*Costa et al., 2021*; *Castelein et al., 2020*). Recently, Sun et al. reported the discovery of a vertebral skeletal stem cell (vSSC) residing in the endplate and marked by expression of the genes *Zic1* and *Pax1*, along with other cell surface markers (*Sun et al., 2023*). These vSSCs also express high levels of *Col11a1* (M Greenblatt, personal communication). It is interesting to consider that AIS-associated variation in collagen α1(XI), perhaps together with mechanical forces, could alter the differentiation trajectory of this cell niche. Altogether, extant data and our results strongly suggest that cell populations at the IVD-vertebral junction region are relevant in AIS pathogenesis. Further investigation is warranted to understand the developmental programs of cells in this region of the spine.

Matrix metalloproteinase 3, also known as stromolysin, is a secreted enzyme expressed in connective tissues and in regions of endochondral ossification (*Ortega et al., 2004*). MMP3 has degradative activity toward a variety of ECM components, including proteoglycans, fibronectin, laminin, but notably not type I collagen (*Sellers and Murphy, 1981*). Additionally, in chondrocytes MMP3 also has been shown to translocate to the nucleus, where it activates transcription of connective tissue growth factor (*CTGF/CCN2*) by binding to an element known as t̲ranscription e̲nhancer d̲ominant i̲n c̲hondrocytes (TRENDIC) (*Eguchi et al., 2008*; *Cui et al., 2017*). Our observations of a *Col11a1-Mmp3* signaling axis in chondrocytes and CEP cells raise the possibility that *Col11a1* variation may have consequences for both MMP3 enzymatic activity levels and *MMP3*-mediated transcriptional programming in these cells. *COL11A1* missense variants, usually altering glycine or proline in Gly-X-Y repeats in the collagen α1(XI) helical domain as with *COL11A1^P1335L*, are reported to be frequent in cutaneous squamous cell carcinomas and have been linked to transcriptional changes and tumor invasiveness (*Lee et al., 2021*). The mechanisms by which chondrocytes or other cells sense such single amino acid changes in collagen α1(XI) and induce subsequent transcriptional responses are unknown but may involve direct interaction with integrins in the pericellular space (*Lee et al., 2021*).

We found that *Col11a1* expression is sensitive to estrogen receptor blockade or knockdown in chondrocytes. Type XI collagen is also a key player in organizing the pericellular space, which is critical for transmitting mechanical forces from the ECM to the cell (*Xu et al., 2016*). Thus, it is interesting to consider that type XI collagen may effectively act as a receptor for environmental cues, i.e., mechanical forces and estrogen signaling, in the adolescent spine. Our study provides new insights into the regulation and signaling role of *Col11a1* in chondrocytes, and it suggests potential mechanisms by which its genetic variation contributes to AIS susceptibility.

## Methods
### Discovery study
The cases in the discovery stage (USA TX: n=1358) were recruited at Scottish Rite for Children. Informed consent to participate in this research was obtained as approved by the Institutional Review Board of the University Texas Southwestern Medical Center, protocol STU 112010-150. Subjects were genotyped on the Illumina HumanCoreExome BeadChip (Illumina, San Diego, CA, USA). For controls, we utilized 12,507 non-AMD GRU (non-age-related macular degeneration general research use) subjects of the European ancestry downloaded from dbGaP website (https://www.ncbi.nlm.nih.gov/gap/) from the International Age-Related Macular Degeneration Genomics Consortium study (IAMDGC: phs001039.v1.p1.c1). The subjects from the IAMDGC study were also genotyped on the Illumina HumanCoreExome Beadchip-24v1.0 platform (*Fritsche et al., 2016*). We merged cases and controls and applied quality controls to the genotypes for 468,801 overlapping SNPs using PLINK.1.9 (*Chang et al., 2015*) as described in *Khanshour et al., 2018*. In summary, samples with sex inconsistencies or from duplicated or related individuals or ancestral outliers as identified by principal component analysis (PCA) were removed, leaving 13,865 samples in the analysis. Genotypes were corrected for strand direction, and SNPs with call-rate per marker <95%, deviating from Hardy-Weinberg equilibrium (cutoff p-value = $10^{-4}$), or with significant missingness rate between cases and controls (cutoff p-value = $10^{-4}$) were removed, leaving 341,759 SNPs in the analysis. Genotypes for SNPs across autosomal chromosomes were imputed using Minimac3 with the 1000G-Phase3.V.5 reference panel as described in the instructions available from the software website (*Das et al., 2016*). Protein-coding changes were annotated with ANNOVAR using RefSeq-based transcripts (*Wang et al., 2010*). External databases included allele frequencies from gnomAD (*Karczewski et al., 2020*) variant pathogenicity in Clinvar (*Landrum et al., 2018*); CADD scores (*Rentzsch et al., 2019*) GERP scores (*Davydov et al., 2010*), and protein domains in IntroPro (*Blum et al., 2021*). Only bi-allelic common (MAF > 0.01) protein-altering SNPs with imputation quality Rsq ≥ 0.3 within matrisome genes (*Naba et al., 2016*) were included for further analysis. Matrisome genes used can be found in the Molecular Signature Database (MsigDB) (*Subramanian et al., 2005*; *Liberzon et al., 2015*; https://www.gsea-msigdb.org/gsea/msigdb/cards/NABA_MATRISOME). Genetic association for the imputed allele dosages in the discovery cohort (USA TX) was performed in Mach2dat (*Li et al., 2009*) using logistic regression with gender and 10 principal components (PCs) as covariates. The genomic regions of the associated loci were visualized with LocusZoom software (*Pruim et al., 2010*) utilizing linkage disequilibrium information from 1000 Genomes EUR populations.

### Meta-analysis study
For the meta-analysis stage we utilized four cohorts – USA MO: n=2951 (1213 cases and 1738 controls), Swedish-Danish populations (SW-D: n=4627 [1631 cases and 2996 controls]), Japan (JP: n=79,211 [5327 cases and 73,884 controls]), and Hong Kong (HK: n=3103 [990 cases and 2113 controls]) – to check significant candidates from the discovery study. Summary statistics across the discovery study and the four replication cohorts (total N=103,757 [10,519 cases and 93,238 controls]) were combined as previously described (*Khanshour et al., 2018*) using METAL (*Willer et al., 2010*).

### SW-D cohort
All patients provided written informed consent. Patients were recruited according to protocols approved by the institutional review boards in Stockholm (protocol #290/202906, #2009/1124-31/2, #2012/1595-31/2), Lund (protocol #LU 200-95, #LU 280-99, #LU 363-02, #567/2008, #2014/804), and

Southern Denmark (protocol #S-2011002). Genotyping and analyses were performed as described in *Khanshour et al., 2018*; *Ameur et al., 2017*.

## USA MO cohort

Whole exome sequencing data from 1213 unrelated idiopathic scoliosis cases of European ancestry with spinal curvature greater than 10-degree Cobb angle were derived from the Adolescent Idiopathic Scoliosis 1000 Exomes Study (dbGAP accession number: phs001677), and patients recruited from St. Louis Children's Hospital, and St. Louis Shriners Hospital for Children. Patients and/or parents provided consent to participate in the study, and IRB approval was obtained from Washington University (protocol #201102118). For controls, exome data from 1738 unrelated samples of European ancestry were provided by investigators at Washington University School of Medicine in St. Louis, MO (dbGAP accession numbers: phs000572.v8.p4 and phs000101.v5.p1), and Oregon Health & Science University in Portland, OR (https://gemini.conradlab.org/). Exome data were aligned to the human genome reference (GRCh37) using BWA-MEM (v0.7.15). Variant calling of single nucleotide variants and insertion and deletion variants were generated first for each single sample in cases and controls and then combining all samples with joint genotyping method, described in GATK Best-Practices (Genome Analysis Toolkit [GATK v3.5] https://gatk.broadinstitute.org/hc/en-us/sections/360007226651-Best-Practices-Workflows). All cases and controls samples were classified as unrelated and of European ancestry using relationship inference (*Manichaikul et al., 2010*) and PCA (*Chang et al., 2015*). Association analysis of variants rs3753841 and rs1042704 were performed using logistic regression adjusting for sex and PCs in PLINK (*Chang et al., 2015*).

## JP cohort

Informed consents were obtained from all the subjects or their parents, and the ethics committee of the Keio University Hospital, Tokyo, approved the study protocol (approved protocol #20080129). 5327 case subjects were recruited from collaborating hospitals (Japanese Scoliosis Clinical Research Group) as previously described (*Kou et al., 2019*). For controls, 73,884 subjects were randomly selected from the BioBank Japan Project, and subjects were genotyped on Illumina Human BeadChips as previously described (*Ogura et al., 2015*, *Kou et al., 2013*). Imputation and association analyses in JP were performed as previously described (*de Klerk et al., 1988*).

## HK cohort

3103 subjects were recruited at The Duchess of Kent Children's Hospital. Informed consent to participate in research was obtained as approved by the Institutional Review Board of the University of Hong Kong/Hospital Authority Hong Kong West Cluster (IRB approval number: UW 08-158). All 990 cases were characterized by Cobb angles greater than 40 degrees with onset age between 10 and 18 years. Congenital, neuromuscular, and syndromic scoliosis samples were excluded. We used 2113 controls from the Chinese population with no spinal deformities on MRI scans (*Song et al., 2013*). Cases and controls were genotyped using the Illumina Infinium OmniZhongHua-8 BeadChip and analyzed with GenomeStudio 2.0 software. The quality control approach adopted the GWA tutorial developed by *Marees et al., 2018*. The filtered genotyping data of cases and controls was phased and imputed using SHAPEIT (*Delaneau et al., 2011*) and IMPUTE2 (*Howie et al., 2009*), respectively. Logistic model association analysis was performed using PLINK 1.9 (*Chang et al., 2015*).

## Stratification-by-sex test

To investigate sex specificity in the *COL11A1* and *MMP14* loci, we performed stratification-by-sex analysis in the discovery study (USA_TX). Association for the imputed allele dosages in rs3753841 and rs1042704 was computed separately for females (1157 cases and 7138 controls) using logistic regression with 10 PCs as covariates in Mach2dat (*Li et al., 2009*).

## RNAseq of human tissues

RNAseq was performed as previously described (*Makki et al., 2021*). Read counting and transcript quantification were performed using HTSeq (*Anders et al., 2015*). Finally, reads were normalized using DESeq2 tools (*Love et al., 2014*) and TPM values were generated using the Kalisto pipeline (*Bray et al., 2016*).

## Animal studies

Mouse and rat work was conducted per IACUC approved protocols at University of Texas Southwestern Medical Center (approved protocol #2016-101455) and University of California San Francisco (approved protocol #AN181381) and was in accordance with AALAC and NIH guidelines.

### Generation of *Pax1* knockout mice

Two gRNAs were designed to target the 5′ and 3′ ends of *Pax1* gene (gRNA sequence shown in *Figure 3—figure supplement 1*) using the gRNA design tool on the Integrated DNA Technologies (IDT, Newark, NJ, USA) website and selected based on low off-target and high on-target scores. The knockout allele was generated using *i*-GONAD (*Gurumurthy et al., 2019*) as previously described (*Ushiki et al., 2021*).

To validate proper generation of the knockout, mice were analyzed by genotyping (with primers shown in Appendix 1), Sanger sequencing of PCR-amplified DNA, and southern blot (*Figure 3—figure supplement 1*). For southern blot analyses, genomic DNA were treated with NcoI (Cat #R0193, New England Biolabs, MA, USA) and fractionated by agarose gel electrophoreses. Following capillary transfer onto nylon membranes, blots were hybridized with digoxigenin (DIG)-labeled DNA probes (corresponding to chr2:147,202,083–147,202,444; mm9) amplified by the PCR DIG Probe Synthesis Kit (Cat #11636090910, Sigma-Aldrich, MO, USA). The hybridized probe was immunodetected with antidigoxigenin Fab fragments conjugated to alkaline phosphatase (Cat #11093274910, Sigma-Aldrich, MO, USA) and visualized with a CDP star (Cat #11685627001, Sigma-Aldrich, MO, USA) according to the manufacturer's protocol. Chemiluminescence was detected using the FluorChem E (Cat #92-14860-00, ProteinSimple, CA, USA).

### *Col11a1*^fl/fl and ATC mice

The *Col11a1*^fl/fl mouse line (*Sun et al., 2020*) was kindly provided by Dr. Lou Soslowsky with permission from Dr. David Birk. ATC mice (*Dy et al., 2012*) were kindly provided by Dr. Ryan Gray, with permission from Dr. Veronique Lefebvre.

### Other mice

Cartilage was harvested from C57B/6 wild-type mice for siRNA-mediated knockdown experiments.

## Histological methods

For thin cryostat sections, P0.5 mouse whole body was fixed in 4% paraformaldehyde (PFA) for 6 hr followed by 10% sucrose for 12 hr, then transferred to 18% sucrose for 24 hr. Tissues were then embedded in optimal cutting temperature compound (OCT) and sectioned using low-profile blades on a Thermo Shandon Cryostar NX70 cryostat and all sections were lifted on APES clean microscope slides. For whole mount images, samples were treated similarly with the addition of 2% polyvinylpyrrolidone (PVP) during the cryoprotection step and frozen in 8% gelatin (porcine) in the presence of 20% sucrose and 2% PVP. Samples were sectioned at a thickness of 10 µm. Slides were stored at –80°C until ready for use. For P28 and older mice, spines were removed then fixed, decalcified, and embedded in OCT. Spines were processed by making 7 µm thick lateral cuts the length of the spine.

Collagen α1(XI) was detected by IHC staining using affinity-purified antisera against peptide (C) YGTMEPYQTETPRR-amide (Genescript, NJ, USA) as described (*Sun et al., 2020*), and secondary horseradish peroxidase (HRP)-conjugated affinity-purified secondary antibody (Cat #AP187P, MilliporeSigma Aldrich, MO, USA). Briefly, frozen sections were equilibrated to room temperature for 1 hr, then fixed with 4% PFA in PBS at 4°C for 20 min. Slides were washed, treated with 3% $H_2O_2$ in methanol for 10 min to block endogenous peroxidase, washed, and transferred to PBS with 0.05% Tween 20 (Cat #P3563-10PAK, Sigma-Aldrich, MO, USA) pH 7.4. Slides were blocked with 0.5% goat serum in PBS mix with 0.2% Triton 100 (Cat #T8787, Sigma-Aldrich, MO, USA) at room temperature for 1.5 hr. The primary collagen α1(XI) affinity-purified antibody was applied at 0.40 mg/ml and slides were incubated overnight at 4°C. Afterward slides were washed in PBS Tween 20 for three times and treated with goat anti-rabbit-HRP for 1.5 hr, then washed three times in PBS Tween 20. After applying 3,3′-diaminobenzidine solution, slides were washed and counterstained with Mayer's hematoxylin (Cat #MHS80, Sigma-Aldrich, MO, USA), washed, dehydrated, and mounted.

For collagen α1(XI) and PAX1 IF studies, P0.5 mice, P28 spine, and ribs sections were fixed in 4% PFA for 20 min then washed with PBS + 0.1% Triton three times, before incubation with 10% normal goat serum in PBS + 0.1% Triton for 30 min to block the background. Slides were incubated with goat anti-mouse collagen α1(XI) antibody at 1:500 dilution and mouse anti-rat PAX1(Cat #MABE1115M, Sigma-Aldrich, MO, USA), in PBS + 0.1% Triton + 1% normal goat serum at 4°C overnight. Secondary antibodies used were 1:5000 anti-rat Alexa488 and anti-mouse Alexa594-conjugated antibodies (Cat #A32740 Invitrogen, CA, USA). The sections were mounted using ProLong Gold with DAPI (Cat #S36964 Invitrogen, CA, USA) for imaging as described (*Yu et al., 2018*). All images were taken with Carl Zeiss Axio Imager.M2 fluorescence microscope (Zeiss, Oberkochen, DE).

## Rib cartilage and IVD cell culture

All cell culture experiments utilized primary cells. Cell cultures were negative for mycoplasma contamination as determined by random, monthly testing. Mouse costal chondrocytes were isolated from the rib cage and sternum of P0.5 mice. Rat IVD was removed intact from 1-month female rats and immediately separated into NP, AF, and CEP isolates. Subsequently, tissues were incubated and shaken with 2 mg/ml Pronase solution (Cat #10165921001 Sigma-Aldrich, Inc, St. Louis, MO, USA) for 1 hr, followed by 1.5 hr digestion with 3 mg/ml Collagenase D solution (Cat #11088882001 Sigma-Aldrich, Inc, St. Louis, MO, USA), then 5 hr digestion with 0.5 mg/ml Collagenase D solution before three times PBS wash. Filtered, dissociated cells were seeded in Dulbecco's modified Eagle's medium (DMEM; Cat #MT15017CV Thermo Fisher Scientific, MA, USA) containing 10% fetal bovine serum (FBS), 100 µg/ml streptomycin, and 100 IU/ml penicillin. Remaining cartilage tissues underwent further digestion in 86 U/ml type 1 collagenase (Cat #SCR103 Sigma-Aldrich, Inc, St. Louis, MO, USA) overnight. Cells were collected and cultured in DMEM with 10% FBS plus 100 µg/ml streptomycin and 100 IU/ml penicillin.

## SV40 immortalization and transfection of primary chondrocytes

*Col11a1<sup>fl/fl</sup>* mouse costal chondrocytes were isolated from the rib cage and sternum of P0.5 mice. The cells were transduced with pRRLsin-sv40 T antigen-IRES-mCherry lentivirus (*Jha et al., 1998*) for 48 hr, then sorted for mCherry-positive cells by flow cytometry. mCherry-positive cells were then infected with plv-eGFP, plv-eGFP-COL11A1-HA, plveGFP-COL11A1<sup>P1335L</sup>-HA constructs. After expansion, GFP-positive cells were sorted by flow cytometry and seeded in 24-well plates.

## Adenovirus treatment

SV40-induced *Col11a1<sup>fl/fl</sup>* mouse costal chondrocytes were grown to 50% confluency. Afterward, cells were treated with 2 µl Ad5-CMV-cre adenovirus (titer $1.8{\times}10^{11}$ pfu/ml) and Ad5-CMV-eGFP (titer $1.65{\times}10^{10}$ pfu/ml) as control. Both virus strains were from the Gene Vector Core facility, Baylor College of Medicine. After 48 hr the cells were harvested for mRNA and protein lysate.

## RNAseq and qRT-PCR

For *Pax1* knockout studies, total RNA was collected from E12.5 tails using TRIzol (Cat #15596026, Thermo Fisher Scientific, MA, USA) and converted to cDNA using ReverTra Ace qPCR-RT master mix with genomic DNA remover (Cat #FSQ-301, Toyobo, Osaka, Japan). Sequencing was done using an Illumina Novaseq platform and the data were analyzed using Partek Flow (version 10.0) and gene ontology (*Ashburner et al., 2000*). qPCR was performed using SsoFast EvaGreen supermix (Cat #1725205, Bio-Rad, CA, USA). Primer sequences used for qPCR are shown in Appendix 1.

To quantify the expression level of *Col11a1*, *Mmp3*, and marker genes in IVD compartments and rib cartilage, cultured cells were collected in RNeasy (QIAGEN, Inc) for RNA purification. Taqman Reverse Transcription Kit (Cat #4387406 Thermo Fisher Scientific, MA, USA) was used to reverse-transcribe mRNA into cDNA. Following this, RT-qPCR was performed using a Power SYBR Green PCR Master Mix Kit (Cat #1725271, Bio-Rad, CA, USA). The primer sequences for the genes used in this study are listed in Appendix 1. Gene expression was calculated using the ΔΔCT method after normalizing to GAPDH.

## siRNA knockdown

Mouse rib cartilage cells seeded in six-well plates were 60–80% confluent at transfection. Lipofect-amine RNAiMAX reagent (Cat #13778030 Thermo Fisher, Waltham, MA, USA) was diluted (9 µl in 500 µl) in Opti-MEM Medium (Cat #31985070 Thermo Fisher, MA, USA). 30 pmol siRNA was diluted in 500 µl Opti-MEM medium, then added to diluted Lipofectamine RNAiMAX Reagent. siRNA-lipid complex was added to cells after 5 min incubation at room temperature. Cells were harvested after 72 hr.

## Western blotting

For MMP3 western blotting, a total of 30 µg protein mixed with SDS-PAGE buffer was loaded on 12% SDS-polyacrylamide gel for electrophoresis. For collagen α1(XI) western blotting, 50 µg protein mixed with SDS-PAGE buffer was loaded on 4–20% SDS-polyacrylamide gel. The separated proteins were then transferred to nitrocellulose membranes (Cat #77010 Thermo Fisher Waltham, MA, USA) at 100 V for 2–3 hr. The membrane was first incubated with blocking buffer containing 5% defatted milk powder, and then exposed to 0.1 mg/ml mouse anti-rabbit Mmp3 (Cat #ab214794 Abcam, Cambridge, MA, USA) or anti-rabbit Col11a1 (Cat #PA5-38888 Thermo Fisher, Waltham, MA, USA) overnight. The samples were then washed thoroughly with TBS buffer, followed by incubation with HRP-labeled anti-rabbit IgG secondary antibodies 1:5000 (Cat #32460 Thermo Fisher, Waltham, MA, USA) overnight. The membranes were then washed with TBS buffer. GAPDH was detected by a rabbit anti-mouse antibody (Cat #14C10 Cell Signaling, MA, USA) and used as the internal loading control.

## Acknowledgements

We thank the patients and their families who participated in these studies. We are grateful to the clinical investigators who referred patients into the study from the Japan Scoliosis Clinical Research Group (JSCRG), and the Scottish Rite for Children Clinical Group (SRCCG). We are grateful for the Scoliosis and Genetics in Scandinavia (ScoliGeneS) study group for help with patient recruitment and analyses of the Swedish-Danish cohort. The names and affiliations for JSCRG, Scottish Rite for Children Clinical Group (SRCCG), ScoliGeneS are included in Appendix 2. We also thank Drs. Carlos Cruchaga, Sanjay Jain, Matthew Harms, and Don Conrad for allowing us to use exome data from their cohort studies. We are grateful to Dr. James Lupski and the Baylor-Hopkins Center for Mendelian Genomics for providing exome sequencing of AIS cases. The University of Pennsylvania is acknowledged for providing the Col11a1$^{fl/fl}$ mouse line. We thank Drs. D Burns for help in interpreting histological studies. We thank Dr. M Hilton for sharing scRNAseq data as described in reference (51). This study was funded by JSPS KAKENHI Grants (22H03207 to SI), the Swedish Research Council (number K-2013–52X-22198-01-3 and 2017-01639 to PG and EE), the regional agreement on medical training and clinical research (ALF) between Stockholm County Council and Karolinska Institutet (to PG), Center for Innovative Medicine (CIMED), Karolinska Institutet (to PG), the Department of Research and Development of Vasternorrland County Council (to PG), and Karolinska Institutet research funds (to PG) and Research Grants Council of Hong Kong (No: 17114519 to YQS), The National Institutes of Health GM127390 (to NG) and the Eunice Kennedy Shriver National Institute of Child Health & Development of the National Institutes of Health (P01HD084387 to CAW and NA). Content is solely the responsibility of the authors and does not necessarily represent the official views of the NIH. The data used for the analyses described in the USA MO cohort were obtained from the database of Genotypes and Phenotypes (dbGaP) Adolescent Idiopathic Scoliosis 1000 Exomes Study (phs001677. v1.p1) which included investigators at Washington University in St. Louis (M Dobbs), Shriners Hospital for Children, St. Louis (M Dobbs), University of Colorado (N Miller), University of Iowa (S Weinstein and J Morcuende), University of Wisconsin (P Giampietro), and Hospital for Special Surgery (C Raggio). Sequencing of these samples was funded by NIH NIAMS 1R01AR067715 (M Dobbs and CG). Use of the dbGAP dataset phs001039.v1.p1 is gratefully acknowledged. All contributing sites and additional funding information for dbGAP dataset phs001039.v1.p1 are acknowledged in this publication: Fritsche et al. (2016) Nature Genetics 48 134–143, (doi:10.1038 /ng.3448); The International AMD Genomics consortium's web page is: http://eaglep.case.edu/iamdgc_web/, and additional information is available on: http://csg.sph.umich.edu/abecasis/public/amd015/. The authors acknowledge the Texas Advanced Computing Center (TACC) at The University of Texas at Austin for providing

computing resources that have contributed to the research results reported within this paper. URL: http://www.tacc.utexas.edu.

## Additional information

### Funding

| Funder | Grant reference number | Author |
| --- | --- | --- |
| Japan Society for the Promotion of Science | 22H03207 | Shiro Ikegawa |
| Swedish Research Council | K-2013-52X-22198-01-3 | Elisabet Einarsdottir |
| Swedish Research Council | 2017-01639 | Elisabet Einarsdottir |
| Regional Agreement on Medical Training and Clinical Research | | Paul Gerdhem |
| Center for Innovative Medicine | | Paul Gerdhem |
| Department of Research and Development of Vasternorrland County Council | | Paul Gerdhem |
| Research Grants Council, University Grants Committee | 17114519 | You-qiang Song |
| National Institutes of Health | GM127390 | Nick V Grishin |
| National Institutes of Health | P01HD084387 | Carol A Wise |
| National Institutes of Health | 1R01AR067715 | Christina A Gurnett |

The funders had no role in study design, data collection and interpretation, or the decision to submit the work for publication.

### Author contributions

Hao Yu, Investigation, Methodology, Writing – original draft; Anas M Khanshour, Data curation, Formal analysis, Writing – original draft; Aki Ushiki, Investigation, Writing – original draft; Nao Otomo, Elisabet Einarsdottir, Yanhui Fan, Lilian Antunes, Yared H Kidane, Formal analysis, Writing – review and editing; Yoshinao Koike, Jimin Pei, Nick V Grishin, Formal analysis; Reuel Cornelia, Bret M Evers, Methodology, Writing – review and editing; Rory R Sheng, Yichi Zhang, Investigation; Jason Pui Yin Cheung, You-qiang Song, Christina A Gurnett, Paul Gerdhem, Validation, Writing – review and editing; John A Herring, Investigation, Writing – review and editing; Chikashi Terao, Shiro Ikegawa, Jonathan J Rios, Supervision, Writing – review and editing; Nadav Ahituv, Supervision, Writing – original draft; Carol A Wise, Conceptualization, Supervision, Writing – original draft, Writing – review and editing

### Author ORCIDs

Aki Ushiki http://orcid.org/0000-0002-0889-4577
Bret M Evers http://orcid.org/0000-0001-5686-0315
Jason Pui Yin Cheung http://orcid.org/0000-0002-7052-0875
Chikashi Terao http://orcid.org/0000-0002-6452-4095
You-qiang Song http://orcid.org/0000-0001-9407-2256
Jonathan J Rios https://orcid.org/0000-0002-0969-2184
Nadav Ahituv http://orcid.org/0000-0002-7434-8144
Carol A Wise https://orcid.org/0000-0002-6790-2194

## Ethics

The cases in the discovery stage (USA TX: n=1,358) were recruited at Scottish Rite for Children. Informed consent to participate in this research was obtained as approved by the Institutional Review Board of the University Texas Southwestern Medical Center, protocol STU 112010-150. Cases in the replication cohort SW-D were recruited according to protocols approved by the institutional review boards in Stockholm (protocol #290/202906, #2009/1124-31/2, #2012/1595-31/2), Lund (protocol #LU 200-95, #LU 280-99, #LU 363-02, #567/2008, #2014/804), and Southern Denmark (protocol #S-2011002). Cases in the replication cohort USA-MO cohort were recruited from St. Louis Children's Hospital, and St. Louis Shriners Hospital for Children. Informed consent to participate in the study was obtained from each case as approved by the Washington University Institutional Review Board (protocol #201102118). Cases in the replication cohort JP provided informed consents as approved by the ethics committee of the Keio University Hospital, Tokyo (approved protocol #20080129). For the replication cohort HK, subjects were recruited at The Duchess of Kent Children's Hospital and provided informed consent to participate in research as approved by the Institutional Review Board of the University of Hong Kong/Hospital Authority Hong Kong West Cluster (Institutional Review Board approval number: UW 08-158).

Mouse and rat work was conducted per IACUC approved protocols at University of Texas Southwestern Medical Center (approved protocol #2016-101455) and University of California San Francisco (approved protocol #AN181381) and was in accordance with AALAC and NIH guidelines.

Reviewer #1 (Public Review): https://doi.org/10.7554/eLife.89762.4.sa1
Reviewer #2 (Public Review): https://doi.org/10.7554/eLife.89762.4.sa2
Reviewer #3 (Public Review): https://doi.org/10.7554/eLife.89762.4.sa3
Author Response https://doi.org/10.7554/eLife.89762.4.sa4

# Additional files

## Supplementary files

- MDAR checklist
- Supplementary file 1. Tests of association with variants in 597 matrisome genes.
- Supplementary file 2. Rare COL11A1 variants detected in 625 AIS exomes.

## Data availability

Summary data for GWAS3 is deposited in the NHGRI-EBI GWAS catalog (accession #GCST006902).

The following previously published datasets were used:

| Author(s) | Year | Dataset title | Dataset URL | Database and Identifier |
|---|---|---|---|---|
| Khanshour AM, Kou I, Fan Y, Einarsdottir E, Makki N, Kidane YH, Kere J, Grauers A, Johnson TA, Paria N, Patel C, Singhania R, Kamiya N, Takeda K, Otomo N, Watanabe K, Luk KDK, Cheung KMC, Herring JA, Rios JJ, Ahituv N, Gerdhem P, Gurnett CA, Song YQ, Ikegawa S, Wise CA | 2018 | Genome-wide meta-analysis and replication studies in multiple ethnicities identify novel adolescent idiopathic scoliosis susceptibility loci | https://www.ebi.ac.uk/gwas/studies/GCST006902 | GWAS catalog, GCST006902 |

*Continued on next page*

*Continued*

| Author(s) | Year | Dataset title | Dataset URL | Database and Identifier |
|---|---|---|---|---|
| Fritsche LG, Igl W, Bailey JN, Grassmann F, Sengupta S, Bragg-Gresham JL, Burdon KP, Hebbring SJ, Wen C, Gorski M, Kim IK, Cho D, Zack D, Souied E, Scholl HP, Bala E, Lee KE, Hunter DJ, Sardell RJ, Mitchell P, Merriam JE, Cipriani V, Hoffman JD, Schick T, Lechanteur YT, Guymer RH, Johnson MP, Jiang Y, Stanton CM, Buitendijk GH, Zhan X, Kwong AM, Boleda A, Brooks M, Gieser L, Ratnapriya R, Branham KE, Foerster JR, Heckenlively JR, Othman MI, Vote BJ, Liang HH, Souzeau E, McAllister IL, Isaacs T, Hall J, Lake S, Mackey DA, Constable IJ, Craig JE, Kitchner TE, Yang Z, Su Z, Luo H, Chen D, Ouyang H, Flagg K, Lin D, Mao G, Ferreyra H , Stark K, von Strachwitz CN, Wolf A, Brandl C, Rudolph G, Olden M, Morrison MA, Morgan DJ, Schu M, Ahn J, Silvestri G, Tsironi EE, Park KH, Farrer LA, Orlin A, Brucker A, Li M, Curcio CA, Mohand-Saïd S, Sahel JA, Audo I, Benchaboune M, Cree AJ, Rennie CA, Goverdhan SV, Grunin M, Hagbi-Levi S, Campochiaro P, Katsanis N, Holz FG, Blond F, Blanché H, Deleuze JF, Igo RP, Truitt B, Peachey NS, Meuer SM, Myers CE, Moore EL, Klein R, Hauser MA, Postel EA, Courtenay MD, Schwartz SG, Kovach JL, Scott WK, Liew G, Tan AG, Gopinath B, Merriam JC, Smith RT, Khan JC, Shahid H, Moore AT, McGrath JA, Laux R, Brantley MA, Agarwal A, Ersoy L, Caramoy A, Langmann T, Saksens NT, de Jong EK, Hoyng CB, Cain MS, Richardson AJ, Martin TM, Blangero J, Weeks DE, Dhillon B, van Duijn CM, Doheny KF, Romm J, Klaver CC, Hayward C, Gorin MB, Klein ML, Baird PN, den Hollander AI, Fauser S, Yates JR, Allikmets R, Wang JJ, Schaumberg DA, Klein BE, Hagstrom SA, Chowers I, Lotery AJ, Léveillard T, Zhang K, Brilliant MH, Hewitt AW, Swaroop A, Chew EY, Pericak-Vance MA, DeAngelis M, Stambolian D, Haines JL, Iyengar SK, Weber BH, Abecasis GR, Heid IM | 2016 | International Age-Related Macular Degeneration Genomics Consortium - Exome Chip Experiment | https://www.ncbi.nlm.nih.gov/projects/gap/cgi-bin/study.cgi?study_id=phs001039.v1.p1 | NCBI, phs001039.v1.p1 |

*Continued on next page*

*Continued*

| Author(s) | Year | Dataset title | Dataset URL | Database and Identifier |
|---|---|---|---|---|
| Gurnett CA, Dobbs MB, Miller N, Morcuende J, Giampietro P, Raggio C, Wise C | 2019 | Adolescent Idiopathic Scoliosis (AIS) 1000 Exomes Study | https://www.ncbi.nlm.nih.gov/projects/gap/cgi-bin/study.cgi?study_id=phs001677.v1.p1&phv=395441&phd=&pha=&pht=8379&phvf=&phdf=&phaf=&phtf=&dssp=1&consent=&temp=1 | NCBI, phs001677.v1.p1 |

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

# Appendix 1

## siRNA and primer sequences

**Appendix 1—table 1.** RNA and DNA oligonucleotide primers used for siRNA knockdown, RT-qPCR, and genotyping experiments.

| | |
|---|---|
| Mouse Esr2 siRNA | CAAGUGUUACGAAGUAGGAdT |
| Mouse Col11a1 siRNA | GAAAGAAGGUGCAAAGGGUdT |
| Mouse Mmp3 F | CTCTGGAACCTGAGACATCACC |
| Mouse Mmp3 R | AGGAGTCCTGAGAGATTTGCGC |
| Mouse Col11a1 F | AGGAGAGTTGAGAATTGGGAATC |
| Mouse Col11a1 R | TGGTGATCAGAATCAGAAGTT |
| Mouse Col11a2 F | CTCATCTTCCTGCATCAGAC |
| Mouse Col11a2 R | ACTTGGAAAGCGAGGTCCT |
| Mouse Adgrg6 F | AGAGGATGGACTGAGGCTGTGT |
| Mouse Adgrg6 R | CCAGGCTTGTTTGGACATGGTTG |
| Mouse Sox6 F | GCATAAGTGACCGTTTTGGCAGG |
| Mouse Sox6 R | GGCATCTTTGCTCCAGGTGACA |
| Mouse Mmp14 F | GCCTTCTGTTCCTGATAA |
| Mouse Mmp14 R | CCATCCTTCCTCTCGTAG |
| Mouse Pax1 F | AACCAGCACGGAGTATACAGC |
| Mouse Pax1 R | TGTAAGCTACCGAGTGCATCC |
| Mouse Esr2 F | GGTCCTGTGAAGGATGTAAGGC |
| Mouse Esr2 R | TAACACTTGCGAAGTCGGCAGG |
| Mouse Gapdh F | CATCACTGCCACCCAGAAGACTG |
| Mouse Gapdh R | ATGCCAGTGAGCTTCCCGTTCAG |
| Rat Sfrp2 F | CGTGAAACGGTGGCAGAAG |
| Rat Sfrp2 R | CGGATGCTGCGGGAGAT |
| Rat Krt19 F | AAGACACACTGGCAGAAACG |
| Rat Krt19 R | GATTCTGCCGCTCACTATCA |
| Rat Mmp12 F | TTGGCCATTCCTTGGGGCTGC |
| Rat Mmp12 R | TGTTGGTGGCTGGACTCCCAGG |
| Mouse Pax1 F (*Figure 5*) | CCGCACATTCAGTCAGCAAC |
| Mouse Pax1 R (*Figure 5*) | CATCTTGGGGGAGTAGGCAG |
| Mouse Col11a1 F (*Figure 5*) | CACAAAACCCCTCGATAGAAGTG |
| Mouse Col11a1 R (*Figure 5*) | CCTGTGATCAGGAACTGCTGAA |
| Mouse Adgrg6 F (*Figure 5*) | TCCTGTCCATCTCTGGCTCA |
| Mouse Adgrg6 R (*Figure 5*) | CACAAGACAGAGCTGCTCCA |
| Mouse Sox6 F (*Figure 5*) | TGCGACAGTTCTTCACTGTGG |
| Mouse Sox6 R (*Figure 5*) | CGTCCATCTTCATACCATACG |
| Mouse β-Actin F (*Figure 5*) | GGCACCACACCTTCTACAATG |
| Mouse β-Actin R (*Figure 5*) | GGGGTGTTGAAGGTCTCAAAC |

*Appendix 1—table 1 Continued on next page*

*Appendix 1—table 1 Continued*

| Pax1-genotyping F | CAGAACCTGGAATGCTGTGCTC |
|---|---|
| Pax1-genotyping R | AAAGGGTTGCAGTGCCTTCAC |

## Appendix 2

### Clinical groups

Scottish Rite for Children Clinical Group

Lori A Karol[1], Karl E Rathjen[1], Daniel J Sucato[1], John G Birch[1], Charles E Johnston III[1], Benjamin S Richards[1], Brandon Ramo[1], Amy L McIntosh[1], John A Herring[1], Todd A Milbrandt[2], Vishwas R Talwakar[3], Henry J Iwinski[3], Ryan D Muchow[3], J Channing Tassone[4], X-C Liu[4], Richard Shindell[5], William Schrader[6], Craig Eberson[7], Anthony Lapinsky[8], Randall Loder[9], and Joseph Davey[10]

1. Department of Orthopaedic Surgery, Texas Scottish Rite Hospital for Children, Dallas, Texas, USA
2. Department of Orthopaedic Surgery, Mayo Clinic, Rochester, Minnesota, USA
3. Department of Orthopaedic Surgery, Shriners Hospitals for Children, Lexington, Kentucky, USA
4. Department of Orthopaedic Surgery, Children's Hospital of Wisconsin, Milwaukee, Wisconsin, USA
5. OrthoArizona, Phoenix, Arizona, USA
6. Departments of Orthopedics, Sports Medicine, and Surgical Services, Akron Children's Hospital, Akron, Ohio, USA
7. Pediatric Orthopaedics and Scoliosis, Hasbro Children's Hospital, Providence, Rhode Island, USA
8. University of Massachusetts Memorial Medical Center, Worcester, Massachusetts, USA
9. Indiana University-Purdue University Indianapolis, Indianapolis, Indiana, USA
10. University of Oklahoma Health Sciences Center, Oklahoma City, Oklahoma, USA

Japan Scoliosis Clinical Research Group

Kota Watanabe[1], Nao Otomo[1,2], Kazuki Takeda[1,2], Yoshiro Yonezawa[1,2], Yoji Ogura[1,2], Yohei Takahashi[1,2], Noriaki Kawakami[3], Taichi Tsuji[4], Koki Uno[5], Teppei Suzuki[5], Manabu Ito[6], Shohei Minami[7], Toshiaki Kotani[7], Tsuyoshi Sakuma[7], Haruhisa Yanagida[8], Hiroshi Taneichi[9], Satoshi Inami[9], Ikuho Yonezawa[10], Hideki Sudo[11], Kazuhiro Chiba[12], Naobumi Hosogane[12], Kotaro Nishida[13], Kenichiro Kakutani[13], Tsutomu Akazawa[14], Takashi Kaito[15], Kei Watanabe[16], Katsumi Harimaya[17], Yuki Taniguchi[18], Hideki Shigemats[19], Satoru Demura[20], Takahiro Iida[21], Ryo Sugawara[22], Katsuki Kono[23], Masahiko Takahata[24], Norimasa Iwasaki[24], Eijiro Okada[1], Nobuyuki Fujita[1], Mitsuru Yagi[1], Masaya Nakamura[1], Morio Matsumoto[1]

1. Department of Orthopaedic Surgery, Keio University School of Medicine, Tokyo, Japan
2. Laboratory of Bone and Joint Diseases, Center for Integrative Medical Sciences, RIKEN, Tokyo, Japan
3. Department of Orthopaedic Surgery, Meijo Hospital, Nagoya, Japan
4. Department of Orthopaedic Surgery, Toyota Kosei Hospital, Nagoya, Japan
5. Department of Orthopaedic Surgery, National Hospital Organization, Kobe Medical Center, Kobe, Japan
6. Department of Orthopaedic Surgery, National Hospital Organization, Hokkaido Medical Center, Sapporo, Japan
7. Department of Orthopaedic Surgery, Seirei Sakura Citizen Hospital, Sakura, Japan
8. Department of Orthopaedic Surgery, Fukuoka Children's Hospital, Fukuoka, Japan
9. Department of Orthopaedic Surgery, Dokkyo Medical University School of Medicine, Mibu, Japan
10. Department of Orthopaedic Surgery, Juntendo University School of Medicine, Tokyo, Japan
11. Department of Advanced Medicine for Spine and Spinal Cord Disorders, Hokkaido University Graduate School of Medicine, Sapporo, Japan
12. Department of Orthopaedic Surgery, National Defense Medical College, Tokorozawa, Japan
13. Department of Orthopaedic Surgery, Kobe University Graduate School of Medicine, Kobe, Japan
14. Department of Orthopaedic Surgery, St. Marianna University School of Medicine, Kawasaki, Japan
15. Department of Orthopaedic Surgery, Osaka University Graduate School of Medicine, Suita, Japan
16. Department of Orthopaedic Surgery, Niigata University Hospital, Niigata, Japan
17. Department of Orthopaedic Surgery, Graduate School of Medical Sciences, Kyushu University Beppu Hospital, Fukuoka, Japan
18. Department of Orthopaedic Surgery, Faculty of Medicine, The University of Tokyo, Tokyo, Japan
19. Department of Orthopaedic Surgery, Nara Medical University, Nara, Japan

20. Department of Orthopaedic Surgery, Kanazawa University School of Medicine, Kanazawa, Japan
21. Department of Orthopaedic Surgery, Dokkyo Medical University Koshigaya Hospital, Koshigaya, Japan
22. Department of Orthopaedic Surgery, Jichi Medical University, Simotsuke, Japan
23. Department of Orthopaedic Surgery, Kono Othopaedic Clinic, Tokyo, Japan
24. Department of Orthopaedic Surgery, Hokkaido University, Sapporo, Japan

## Scoliosis and Genetics in Scandinavia study group
Tian Cheng[1], Juha Kere[2], Aina Danielsson[3], Kristina Åkesson[4], Ane Simony[5], Mikkel Andersen[5], Steen Bach Christensen[5], Maria Wikzén[6], Luigi Belcastro[6]

1. Department of Clinical Sciences and Technology, Karolinska Institutet, Huddinge, Sweden
2. Department of Biosciences and Nutrition, Karolinska Institutet, Huddinge, Sweden
3. Department of Orthopedics, Sahlgrenska University Hospital, Gothenburg, Sweden
4. Department of Orthopedics and Clinical Sciences, Lund University, Skane University Hospital, Malmö, Sweden
5. Sector for Spine Surgery and Research, Middelfart Hospital, Middelfart, Denmark
6. Department of Reconstructive Orthopaedics, Karolinska University Hospital, Stockholm, Sweden

