## [Editor Report · eLife assessment]

This **valuable** study analyzes a large cohort of Adolescent Idiopathic Scoliosis (AIS) patients, identifying an association with a variant in COL11A1 (Pro1335Leu). Experimental testing of this potentially pathogenic variant in vitro suggests a connection between Pax1, Col11a1, Mmp3, and estrogen signaling, thus providing **solid** support for the proposed link between hormonal and matrix components in the development of AIS.

---

## [Referee Report · Reviewer #1 (Public Review)]

Summary:

This revised study follows up on previous work showing a female-specific enhancer region of PAX1 is associated with adolescent idiopathic scoliosis (AIS). This new analysis combines human GWAS analysis from multiple countries to identify a new AIS-associated coding variant in the COL11A1 gene (COL11A1P1335L). Using a Pax1 knockout mouse they go on to find that PAX1 and Collagen XI protein are expressed in the intervertebral discs (IVDs) and robustly in the growth plate, showing that COL11A1 expression is reduced in Pax1 mutant growth plate. Moreover, other AIS-associated genes, Gpr126 and Sox6, were also reduced in Pax1 mutant mice, suggesting a common pathway is involved in AIS.

Using SV40 immortalized costal cartilage cells, derived from floxed Col11a1 mice primary rib cage cartilage, they go to show that removal of Col11a1 leads to reduction of Mmp3 expression. In this context, the expression of wild-type Col11a1 restored regular levels of Mmp3 expression, while expression of the AIS-associated Col11a1P1335L allele failed to restore normal Mmp3 expression. This supports a model that the AIS-associated Col11a1P1335L allele leads to the dysregulation of ECM in vivo.

Using this culture system, they go on to test the role of the estrogen receptor ESR2, showing that loss of this receptor leads to reduced Mmp3 and Pax1 expression, and increased Col11a1 expression. They support this by showing similar gene expression changes and estrogen receptor function in Rat cartilage endplate cell culture.

Altogether, this study nicely brings together an impressive number of human genetic data from multi-ethnic AIS cohorts and controls from across the globe and functionally tests these findings in cell culture and animal models. This study wonderfully integrates other findings from other human and mouse work in AIS and supports a new molecular mechanism by which estrogen can interact and synergize with COL11A1/PAX1/MMP3 signaling to change ECM development and dynamics, thus providing a tangible model for mutations and dysregulation of this pathway can increase the susceptibility of scoliosis.

Strengths:

This work integrates a large cohort of human genetic data from AIS patient and control from diverse ethnic backgrounds, across the globe. This work attempts to functionally test their findings in vivio and by use of cell culture.

Weaknesses:

Many of the main functional work was done in cell culture and not in vivo.

---

## [Referee Report · Reviewer #2 (Public Review)]

Summary:

In this manuscript, Yu and colleagues sought to identify new susceptibility genes for adolescent idiopathic scoliosis (AIS). The significance of this work is high, especially given the still large knowledge gap of the mechanistic underpinnings for AIS. In this multidisciplinary body of work, the authors first performed a genetic association study of AIS case-control cohorts (combined 9,161 cases and 80,731 controls) which leveraged common SNPs in 1027 previously defined matrisome genes. Two nonsynonymous variants were found to be significantly associated with AIS: MMP14 p.Asp273Asn and COL11A1 p.Pro1153Leu, the latter of which had the more robust association and remained significant when females were tested independent of males. Next, the authors followed a series of functional validation experiments to support biological involvement of COL11A1 p.Pro1153Leu in AIS through expression, biochemical, and histological studies in physiologically relevant cell and mouse models. Together, the authors propose a hitherto unreported model for AIS that involves the interplay of the COL11A1 susceptibility locus with estrogen signaling to alter a Pax1-Col11a1-Mmp3 signaling axis at the growth plate.

Strengths:

The manuscript is clearly written and follows a series of logical steps toward connecting multiple matrisome genes and putative AIS effectors in a new framework of pathomechanism. The multidisciplinary nature of the work makes it a strong body of work wherein multiple models offer multiple lines of supportive data. Thus, this manuscript remains an important multidisciplinary study of the genetic and functional basis of adolescent idiopathic scoliosis (AIS). To the benefit of the overall manuscript quality, the reviewers have addressed most concerns to satisfaction. Please include the list of three rare missense variants mentioned in the response to reviewers as a supplementary table. Please also include methods for the SKATO rare variant burden analysis.

---

## [Referee Report · Reviewer #3 (Public Review)]

Summary:

This article demonstrates a Pax1-Col11a1-Mmp3 signaling axis associated with adolescent idiopathic scoliosis and finds that estrogen affects this signaling axis. In addition, the authors have identified a new COL11A1 mutation and verified its effect on the Pax1-Col11a1-Mmp3 axis.

Strengths:

1. Col11a1P1335L is verified in multicenter cohorts with high confidence.

2. The article identified a potential pathogenesis of AIS.

Weaknesses:

The SV40-immortalized cell line established from Col11a1fl/fl mouse rib cartilage was applied in the study, and overexpression system was used to confirm that P1335L variant in COL11A1 perturbs its regulation of MMP3. However, due to the absence of P1335L point mutant mice, it cannot be confirmed whether P1335L can actually cause AIS, and the pathogenicity of this mutation cannot be directly verified.

---

## [Author Response]

The following is the authors’ response to the previous reviews.

We thank the reviewers for truly valuable advice and comments. We have made multiple corrections and revisions to the original pre-print accordingly per the following comments:

1. Pro1153Leu is extremely common in the general population (allele frequency in gnomAD is 0.5). Further discussion is warranted to justify the possibility that this variant contributes to a phenotype documented in 1.5-3% of the population. Is it possible that this variant is tagging other rare SNPs in the COL11A1 locus, and could any of the existing exome sequencing data be mined for rare nonsynonymous variants?One possible avenue for future work is to return to any existing exome sequencing data to query for rare variants at the COL11A1 locus. This should be possible for the USA MO case-control cohort. Any rare nonsynonymous variants identified should then be subjected to mutational burden testing, ideally after functional testing to diminish any noise introduced by rare benign variants in both cases and controls. If there is a significant association of rare variation in AIS cases, then they should consider returning to the other cohorts for targeted COL11A1 gene sequencing or whole exome sequencing (whichever approach is easier/less expensive) to demonstrate replication of the association.

Response: Regarding the genetic association of the common COL11A1 variant rs3753841 (p.(Pro1335Leu)), we do not propose that it is the sole risk variant contributing to the association signal we detected and have clarified this in the manuscript. We concluded that it was worthy of functional testing for reasons described here. Although there were several common variants in the discovery GWAS within and around COL11A1, none were significantly associated with AIS and none were in linkage disequilibrium (R^2^>0.6) with the top SNP rs3753841. We next reviewed rare (MAF<=0.01) coding variants within the COL11A1 LD region of the associated SNP (rs3753841) in 625 available exomes representing 46% of the 1,358 cases from the discovery cohort. The LD block was defined using Haploview based on the 1KG_CEU population. Within the ~41 KB LD region (chr1:103365089- 103406616, GRCh37) we found three rare missense mutations in 6 unrelated individuals, Table below. Two of them (NM_080629.2: c.G4093A:p.A1365T; NM_080629.2:c.G3394A:p.G1132S), from two individuals, are predicted to be deleterious based on CADD and GERP scores and are plausible AIS risk candidates. At this rate we could expect to find only 4-5 individuals with linked rare coding variants in the total cohort of 1,358 which collectively are unlikely to explain the overall association signal we detected. Of course, there also could be deep intronic variants contributing to the association that we would not detect by our methods. However, given this scenario, the relatively high predicted deleteriousness of rs3753841 (CADD = 25.7; GERP=5.75), and its occurrence in a GlyX-Y triplet repeat, we hypothesized that this variant itself could be a risk allele worthy of further investigation.

**Author response table 1. sa4table1:** 

VarID_GRCh37	RsID	Mutation	gnomAD	CADD	GERP	#Cases
1:103377744:C:T	rs151249006	NM_080629.2:c.G4093A:p.A1365T	6.37E-05	25.8	5.42	1
1:103381192:C:A	rs150669855	NM_080629.2:c.G3847T:p.V1283L	0.0007	0.001	-10.9	4
1:103405909:C:T	rs370589018	NM_080629.2:c.G3394A:p.G1132S	6.37E-05	25.2	5.46	1

We also appreciate the reviewer’s suggestion to perform a rare variant burden analysis of COL11A1. We did conduct pilot gene-based analysis in 4534 European ancestry exomes including 797 of our own AIS cases and 3737 controls and tested the burden of rare variants in COL11A1. SKATO P value was not significant (COL11A1_P=0.18), but this could due to lack of power and/or background from rare benign variants that could be screened out using the functional testing we have developed.

1. COL11A1 p.Pro1335Leu is pursued as a direct candidate susceptibility locus, but the functional validation involves both: (a) a complementation assay in mouse GPCs, Figure 5; and (b) cultured rib cartilage cells from Col11a1-Ad5 Cre mice (Figure 4). Please address the following:2A. Is Pro1335Leu a loss of function, gain of function, or dominant negative variant? Further rationale for modeling this change in a Col11a1 loss of function cell line would be helpful.

Response: Regarding functional testing, by knockdown/knockout cell culture experiments, we showed for the first time that Col11a1 negatively regulates Mmp3 expression in cartilage chondrocytes, an AIS-relevant tissue. We then tested the effect of overexpressing the human wt or variant COL11A1 by lentiviral transduction in SV40-transformed chondrocyte cultures. We deleted endogenous mouse Col11a1 by Cre recombination to remove the background of its strong suppressive effects on Mmp3 expression. We acknowledge that Col11a1 missense mutations could confer gain of function or dominant negative effects that would not be revealed in this assay. However as indicated in our original manuscript we have noted that spinal deformity is described in the cho/cho mouse, a Col11a1 loss of function mutant. We also note the recent publication by Rebello et al. showing that missense mutations in Col11a2 associated with congenital scoliosis fail to rescue a vertebral malformation phenotype in a zebrafish col11a2 KO line. Although the connection between AIS and vertebral malformations is not altogether clear, we surmise that loss of the components of collagen type XI disrupt spinal development. in vivo experiments in vertebrate model systems are needed to fully establish the consequences and genetic mechanisms by which COL11A1 variants contribute to an AIS phenotype.

2B. Expression appears to be augmented compared WT in Fig 5B, but there is no direct comparison of WT with variant.

Response: Expression of the mutant (from the lentiviral expression vector) is increased compared to mutant. We observed this effect in repeated experiments. Sequencing confirmed that the mutant and wildtype constructs differed only at the position of the rs3753841 SNP. At this time, we cannot explain the difference in expression levels. Nonetheless, even when the variant COL11A1 is relatively overexpressed it fails to suppress MMP3 expression as observed for the wildtype form.

2C. How do the authors know that their complementation data in Figure 5 are specific?Repetition of this experiment with an alternative common nonsynonymous variant in COL11A1 (such as rs1676486) would be helpful as a comparison with the expectation that it would be similar to WT.

Response: We agree that testing an allelic series throughout COL11A1 could be informative, but we have shifted our resources toward in vivo experiments that we believe will ultimately be more informative for deciphering the mechanistic role of COL11A1 in MMP3 regulation and spine deformity.

2D. The y-axes of histograms in panel A need attention and clarification. What is meant by power? Do you mean fold change?

Response: Power is directly comparable to fold change but allows comparison of absolute expression levels between different genes.

2E. Figure 5: how many technical and biological replicates? Confirm that these are stated throughout the figures.

Response: Thank you for pointing out this oversight. This information has been added throughout.

1. Figure 2: What does the gross anatomy of the IVD look like? Could the authors address this by showing an H&E of an adjacent section of the Fig. 2 A panels?

Response: Panel 2 shows H&E staining. Perhaps the reviewer is referring to the WT and Pax1 KO images in Figure 3? We have now added H&E staining of WT and Pax1 KO IVD as supplemental Figure 3E to clarify the IVD anatomy.

1. Page 9: "Cells within the IVD were negative for Pax1 staining ..." There seems to be specific PAX1 expression in many cells within the IVD, which is concerning if this is indeed a supposed null allele of Pax1. This data seems to support that the allele is not null.

Response: We have now added updated images for the COL11A1 and PAX1 staining to include negative controls in which we omitted primary antibodies. As can be seen, there is faint autofluorescence in the PAX1 negative control that appears to explain the “specific staining” referred to by the reviewer. These images confirm that the allele is truly a null.

1. There is currently a lack of evidence supporting the claim that "Col11a1 is positively regulated by Pax1 in mouse spine and tail". Therefore, it is necessary to conduct further research to determine the direct regulatory role of Pax1 on Col11a1.

Response: We agree with the reviewer and have clarified that Pax1 may have either a direct or indirect role in Col11a1 regulation.

1. There is no data linking loss of COL11A1 function and spine defects in the mouse model. Furthermore, due to the absence of P1335L point mutant mice, it cannot be confirmed whether P1335L can actually cause AIS, and the pathogenicity of this mutation cannot be directly verified. These limitations need to be clearly stated and discussed. A Col11a1 mouse mutant called chondroysplasia (cho), was shown to be perinatal lethal with severe endochondral defects (https://pubmed.ncbi.nlm.nih.gov/4100752/). This information may help contextualize this study.

Response: We partially agree with the reviewer. Spine defects are reported in the cho mouse (for example, please see reference 36 Hafez et al). We appreciate the suggestion to cite the original Seegmiller et al 1971 reference and have added it to the manuscript.

1. A recent article (PMID37462524) reported mutations in COL11A2 associated with AIS and functionally tested in zebrafish. That study should be cited and discussed as it is directly relevant for this manuscript.

Response: We agree with the reviewer that this study provides important information supporting loss of function I type XI collagen in spinal deformity. Language to this effect has been added to the manuscript and this study is now cited in the paper.

1. Please reconcile the following result on page 10 of the results: "Interestingly, the AISassociated gene Adgrg6 was amongst the most significantly dysregulated genes in the RNA-seq analysis (Figure 3c). By qRT-PCR analysis, expression of Col11a1, Adgrg6, and Sox6 were significantly reduced in female and male Pax1-/- mice compared to wild-type mice (Figure 3d-g)." In Figure 3f, the downregulation of Adgrg6 appears to be modest so how can it possibly be highlighted as one of the most significantly downregulated transcripts in the RNAseq data?

Response: By “significant” we were referring to the P-value significance in RNAseq analysis, not in absolute change in expression. This language was clearly confusing, and we have removed it from the manuscript.

1. It is incorrect to refer to the primary cell culture work as growth plate chondrocytes (GPCs), instead, these are primary costal chondrocyte cultures. These primary cultures have a mixture of chondrocytes at differing levels of differentiation, which may change differentiation status during the culturing on plastic. In sum, these cells are at best chondrocytes, and not specifically growth plate chondrocytes. This needs to be corrected in the abstract and throughout the manuscript. Moreover, on page 11 these cells are referred to as costal cartilage, which is confusing to the reader.

Response: Thank you for pointing out these inconsistencies. We have changed the manuscript to say “costal chondrocytes” throughout.

Minor pointsOn 10 of the Results: "These data support a mechanistic link between Pax1 and Col11a1, and the AIS-associated genes Gpr126 and Sox6, in affected tissue of the developing tail." qRT-PCR validation of Sox6, although significant, appears to be very modestly downregulated in KO. Please soften this statement in the text.

Response: We have softened this statement.

Have you got any information about how the immortalized (SV40) costal cartilage affected chondrogenic differentiation? The expression of SV40 seemed to stimulate Mmp13 expression. Do these cells still make cartilage nodules? Some feedback on this process and how it affects the nature of the culture what be appreciated.

Response: The “+ or –“ in Figure 5 refers to Ad5-cre. Each experiment was performed in SV40-immortalized costal chondrocytes. We have removed SV40 from the figure and have clarified the legend to say “qRT-PCR of human COL11A1 and endogenous mouse Mmp3 in SV40 immortalized mouse costal chondrocytes transduced with the lentiviral vector only (lanes 1,2), human WT COL11A1 (lane 3), or COL11A1P1335L. Otherwise we absolutely agree that understanding Mmp13 regulation during chondrocyte differentiation is important. We plan to study this using in vivo systems.

Figure 1: is the average Odds ratio, can this be stated in the figure legend?

Response: We are not sure what is being asked here. The “combined odds ratio” is calculated as a weighted average of the log of the odds.

A more consistent use of established nomenclature for mouse versus human genes and proteins is needed.Human:GENE/PROTEINMouse: Gene/PROTEIN

Response: Thank you for pointing this out. The nomenclature has been corrected throughtout the manuscript.

There is no Figure 5c, but a reference to results in the main text. Please reconcile. -There is no Figure 5-figure supplement 5a, but there is a reference to it in the main text.Please reconcile.

Response: Figure references have been corrected.

Please indicate dilutions of all antibodies used when listed in the methods.

Response: Antibody dilutions have been added where missing.

On page 25, there is a partial sentence missing information in the Histologic methods; "#S36964 Invitrogen, CA, USA. All images were taken..."

Response: We apologize for the error. It has been removed.

Table 1: please define all acronyms, including cohort names.

Response: We apologize for the oversight. The legend to the Table has been updated with definitions of all acronyms.

Figure 2: Indicate that blue staining is DAPI in panel B. Clarify that "-ab" as an abbreviation is primary antibody negative.

Response: A color code for DAPI and COL11A1 staining has been added and “-ab” is now defined.

Page 4: ADGRG6 (also known as GPR126)...the authors set this up for ADGRG6 but then use GPR126 in the manuscript, which is confusing. For clarity, please use the gene name Adgrg6 consistently, rather than alternating with Gpr126.

Response: Thank you for pointing this out. GPR126 has now been changed to ADGRG6 thoughout the manuscript.

REF 4: Richards, B.S., Sucato, D.J., Johnston C.E. Scoliosis, (Elsevier, 2020). Is this a book, can you provide more clarity in the Reference listing?

Response: Thank you for pointing this out. This reference has been corrected.

While isolation was addressed, the methods for culturing Rat cartilage endplate and costal chondrocytes are poorly described and should be given more text.

Response: Details about the cartilage endplate and costal chondrocyte isolation and culture have been added to the Methods.

Page 11: 1st paragraph, last sentence "These results suggest that Mmp3 expression"... this sentence needs attention. As written, I am not clear what the authors are trying to say.

Response: This sentence has been clarified and now reads “These results suggest that Mmp3 expression is negatively regulated by Col11a1 in mouse costal chondrocytes.”

Page 13: line 4 from the bottom, "ECM-clearing"? This is confusing do you mean ECM degrading?

Response: Yes and thank you. We have changed to “ECM-degrading”.

Please use version numbers for RefSeq IDs: e.g. NM_080629.3 instead of NM_080629

Response: This change has been made in the revised manuscript.

It would be helpful for readers if the ethnicity of the discovery case cohort was clearly stated as European ancestry in the Results main text.

Response: “European ancestry” has been added at first description of the discovery cohort in the manuscript.

Avoid using the term "mutation" and use "variant" instead.

Response: Thank you for pointing this out. “Variant” is now used throughout the manuscript.

Define error bars for all bar charts throughout and include individual data points overlaid onto bars.

Response: Thank you. Error bars are now clarified in the Figure legends.